# Performance and process-based evaluation of the BARPA-R Australasian regional climate model version 1

Emma Howard[1], Chun-Hsu Su[1], Christian Stassen[1], Rajashree Naha[1,3], Harvey Ye[1], Acacia Pepler[1], Samuel S. Bell[1], Andrew J. Dowdy[1], Simon O. Tucker[2], and Charmaine Franklin[1]

[1]Bureau of Meteorology, Australia
[2]UK Met Office, Exeter, United Kingdom
[3]Monash University, Melbourne, Australia

**Correspondence:** Emma Howard (emma.howard@bom.gov.au)

**Abstract.** Anthropogenic climate change is changing the earth system processes that control the characteristics of natural hazards both globally and across Australia. Model projections of hazards under future climate change are necessary for effective adaptation. This paper presents BARPA-R (the Bureau of Meteorology Atmospheric Regional Projections for Australia), a regional climate model designed to downscale climate projections over the Australasian region with the purpose to investigate future hazards. BARPA-R, a limited area model, has a 17 km horizontal grid-spacing and makes use of the Met Office Unified Model (MetUM) atmospheric model and the Joint UK Land Environment Simulator (JULES) land surface model. To establish credibility and in compliance with the Coordinated Regional Climate Downscaling Experiment (CORDEX) experiment design, the BARPA-R framework has been used to downscale ERA5 reanalysis. Here, an assessment of this evaluation experiment is provided. Performance-based evaluation results are benchmarked against ERA5, with comparable performance between the free-running BARPA-R simulations and observationally constrained reanalysis interpreted as a good result. First, an examination of BARPA-R's representation of Australia's surface air temperature, precipitation and 10-m winds finds good performance overall, with biases including a 1°C cold bias in daily maximum temperatures, reduced diurnal temperature range, and wet biases up to 25 mm/month in inland Australia. Recent trends in daily maximum temperatures are consistent with observational products, while trends in minimum temperatures show overestimated warming and trends in precipitation show underestimated wetting in northern Australia. Precipitation and temperature teleconnections are effectively represented in BARPA-R when present in the driving boundary conditions, while 10-metre winds are improved over ERA5 in six out of eight of the Australian regions considered. The second section of the paper considers the representation of large-scale atmospheric circulation features and weather systems. While generally well represented, convection-related features such as tropical cyclones, the SPCZ, Northwest Cloud-Bands and the monsoon westerlies show more divergence from observations and internal interannual variability than mid-latitude phenomena such as the westerly jets and extra-tropical cyclones. Having simulated a realistic Australasian climate, the BARPA-R framework will be used to downscale two climate change scenarios from seven CMIP6 GCMs.

## 1 Introduction

Australia experiences some of the highest global levels of interannual climate variability. As such, climate hazards are a key risk in Australia, encompassing wildfires (known as 'bushfires' in Australia), high intensity precipitation, tropical and extratropical storms, flooding, heatwaves and drought. The risks associated with climate hazards are already changing as the planet warms and will continue to do so into the future. These hazards are set by a range of factors including the interaction of weather processes across the Australian geography and length-scales from kilometres to hundreds of kilometres. Therefore, climate projections encompassing these scales across Australia are needed to inform the assessment of future natural hazards and associated disaster risk (Binskin et al., 2020).

Projections of hazards in Australia's climate can be sourced from dynamically down-scaled climate projections: powerful tools which can help translate global climate projections to hazard-relevant length-scales (Coppola et al., 2021). These projections are generated by Regional Climate Models, (RCMs), a class of climate models that focus on the simulation of a limited regional domain, rather than the whole globe. Typically, RCMs are Limited Area Models (LAMs) with lateral boundaries sourced from global models, however complementary Stretched Grid Models (SGMs) such as the Conformal Cubic Atmospheric Model (CCAM) can also be used for this purpose (McGregor and Dix, 2008, 2005).

RCMs have been used to study hazard projections across Australia (Herold et al., 2021) with focused studies examining changes in bushfires (Dowdy et al., 2019; Di Virgilio et al., 2019a), East Coast Lows and Extratropical Cyclones (Pepler et al., 2016; Pepler and Dowdy, 2022) and heatwaves (Perkins-Kirkpatrick et al., 2016; Hirsch et al., 2019), extreme precipitation (Bao et al., 2017) amongst others. State-based regional climate projections have been produced to assess the risks associated with a changing climate on a sub-national scale (e.g. Corney et al., 2010; Evans et al., 2014; Clarke et al., 2019; Trancoso et al., 2020).

Due to a wide range of combinations of global projections, emissions pathways, RCMs and downscaling domains that are possible, coordination across different institutions is crucial to ensure that climate information available to users is consistent and comparable (Giorgi et al., 2009). The Coordinated Regional Climate Downscaling Experiment (CORDEX) project is an initiative of the World Climate Research Programme (WCRP) that provides a consistent framework to produce downscaled climate projections (Jones et al., 2011). Global driving model projections for CORDEX are sourced from the Coupled Model Intercomparison Project (CMIP). Due to computational expense, the full CMIP ensemble can generally not be downscaled, and a representative subsample may be coordinated instead at a regional level (e.g. Grose et al., 2023). CORDEX has defined a set of 16 climate regions, including the Australasian region, which consists of Australia, New Zealand, the West Pacific and parts of Southeast Asia (shown in red in Figure 1). Six dynamical RCMs, produced using five independent modelling frameworks, contributed downscaled projections of the Australasian region to the first Coordinated Regional Climate Downscaling Experiment (CORDEX-CMIP5) (Di Virgilio et al., 2019a; Evans et al., 2021).

When downscaling ERA-Interim reanalyses, the CORDEX-Australasia CMIP5 ensemble framework featured persistent cold daily maximum biases of order 2-5 ° C, reduced diurnal temperature ranges and dual-signed precipitation biases with magnitudes up to 40 mm port month (Di Virgilio et al., 2019b). Downscaling of the CMIP5 historical experiment model ensemble

reflected these temperature biases and showed dry precipitation biases in the tropical monsoonal regions and wet biases elsewhere (Evans et al., 2021). However, Evans et al. (2020) showed that the CORDEX-CMIP5 Australasia ensemble generally outperformed the driving GCM ensemble, particularly at simulating the tails of temperature and precipitation distributions.

Here, we introduce the Bureau of Meteorology Atmospheric Regional Projections for Australia (BARPA-R), a RCM designed for the Australasian region. BARPA-R is being developed by the Australian Bureau of Meteorology (henceforth the Bureau) and the Australian Climate Service (ACS), together with a forthcoming convection permitting model, BARPA-C. The BARPA-R model configuration and developmental trials were presented by Su et al. (2022b). This model is a continuation of prototype work developed for the Energy Sector Climate Information (ESCI) project, documented by Su et al. (2021) and
hereon referred to as ESCI-BARPA. BARPA-R adheres to the principle of seamless weather and climate prediction by following the Australian Community Climate and Earth-System Simulator (ACCESS) modelling framework and uses a 17km (0.1545 degree) grid spacing. This means that BARPA-R uses an atmospheric model configuration that is complementary to the Bureau's operational numerical weather prediction (NWP) configuration and seasonal prediction configuration, allowing learnings and developments from NWP to be applied over longer time scales into the regional climate change space. Furthermore, BARPA-R
is being developed in tandem with BARRA2 reanalysis (version 2 of the Bureau of Meteorology high-resolution Atmospheric Regional Reanalysis for Australia, (Su et al., 2022a)), allowing for seamless comparison between the data-assimilated and fully model-based simulations.

The Bureau intends to downscale an ensemble of at least 7 CMIP6 global climate projections (GCMs) using the BARPA-R framework. Downscaling GCMs have been selected based on their performance over Australia, representation of climate
drivers, modelling centre independence and the overall ensemble coverage of a range of warming and precipitation change scenarios in the Australian region, following Grose et al. (2023). Through ACS, BARPA-R is intended to produce complementary regional climate projections to existing Australian RCM systems, broadening the ensemble of climate hazard projections available in the Australasian region. BARPA-R will be compliant with next generation of CORDEX, CORDEX-CMIP6. Since the atmospheric component of ACCESS and the UK Met Office's Unified Model (MetUM) are co-developed and share a code
base, BARPA-R also joins a family of MetUM-based regional climate simulations around the world. These include the PRECIS regional climate modelling system, CP4Africa (Stratton et al., 2018), and the HadREM CORDEX-Europe (Tucker et al., 2022) simulations.

This paper presents an assessment of the BARPA-R evaluation simulation. The evaluation simulation is driven at the lateral boundaries using ERA5 reanalysis (Hersbach et al., 2020) and is designed to test the performance of the RCM. This paper
proceeds as follows. In section 2, description of the BARPA-R model configuration, the evaluation methodology and the reference datasets are provided. Section 3 evaluates the performance of BARPA-R in simulating the observed precipitation and temperature and near-surface wind climates in the Australian region. Section 4 provides a process-based evaluation in order to assess the representation of key circulation features and weather systems in the Australian region.

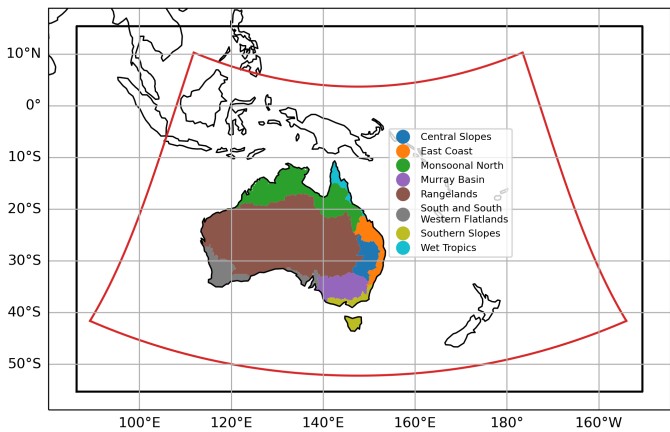

**Figure 1.** Map of region of interest with BARPA-R domain marked by a black box and CORDEX-Australasia domain marked by a red box. The National Resource Management (NRM) clusters described in Section 2.3 and used in model evaluation are indicated in colours as per the figure legend.

## 2 Data and Methods

### 2.1 Experimental Design and Model Configuration

BARPA-R is a land-atmosphere limited area regional climate model. The experimental design follows the CORDEX-v2 Australasia guidelines. The limited area domain covers the CORDEX-Australasia domain, as shown in Figure 1, and includes Australia, New Zealand, the West Pacific and the Maritime Continent. The horizontal grid spacing is 0.1545° of latitude and longitude, which roughly corresponds to 17 km in each direction. 63 vertical model levels have been used, with a 40 km model top. A stretch sigma grid is used with a higher density of levels near the surface, with the first model level is located 10 m Above Ground Level (AGL). Model levels are terrain following near the surface and relax to surfaces of uniform radial height approximately 18 km AGL. The model integration updates on a 7.5 minute dynamical timestep.

The simulation was initialised from the deterministic ERA5 reanalysis on the first of January 1979. Soil moisture was initialised from the January-1 climatological mean of the BARRA-V1 reanalysis (Su et al., 2019). Boundary conditions were updated every 3-hours and derived from the ERA5 pressure level dataset, which consists of 37 vertical levels. The 3D model inputs from ERA5 at the lateral boundaries were horizontal winds, specific humidity, temperature, cloud liquid, cloud ice and cloud cover. Between 2000 and 2006, boundary inputs were derived from ERA5.1 to avoid stratospheric temperature and humidity biases present in the original ERA5 dataset. Sea surface temperatures were sourced from ERA5 and updated daily. Model configuration followed the MetUM standard configuration HadREM3-GA7.05 (Tucker et al., 2022) with a few modifications as described in the BARPA-R version 1 model description paper (Su et al., 2022b). Firstly, the 'fountain buster' correction to the advection scheme was applied to improve moisture conservation during strong convective events. Secondly, the 'prognostic entrainment' scheme (Willet and Whitall, 2017) was applied to improve the representation of convection and

precipitation. Thirdly, Newtonian relaxation (Telford et al., 2008; Stassen et al., 2023) is used to improve alignment between the driving model and the interior of the domain. This relaxation is applied from model level 38 and above (11 km AGL) with a 6-hour relaxation time-scale. These modifications were demonstrated in trial experiments to improve the climatologies of Australian precipitation and near-surface temperatures. The UM and JULES code branches used in the publication have not all been submitted for code review and inclusion in the UM/JULES trunk or released for general use. These branches are associated with nudging, support for the 365-day calendars used by some GCMs, and performance optimisation for the Australian national computational infrastructure (NCI) and were provided to the reviewers of this article.

HadREM3-GA7.05 uses a non-hydrostatic, fully compressible, deep atmosphere formulation with an iterative, semi-implicit dynamical solver (Wood et al., 2014). Awakara-C grid staggering is used in the horizontal dimensions (Arakawa and Lamb, 1977) and Charney-Phillips staggering is used in the vertical dimensions. Key parameterisation schemes include the prognostic condensate (PC2) cloud scheme (Wilson et al., 2008), the Lock et al. (2000) boundary layer scheme, the Gregory and Rowntree (1990) mass flux convection scheme, the Edwards and Slingo (1996) radiation scheme and the Wilson and Ballard (1999) mixed-phase cloud microphysics. These schemes have been routinely improved since their publication through regular model development (Walters et al., 2019). Observed historical green-house gas, aerosol and ozone forcing are implemented following Tucker et al. (2022). This approach prescribes 4D aerosol optical properties on 9 shortwave and 6 longwave bands in the SOCRATES radiative transfer code, combining seasonal and spatial variation derived from an offline simulation using the Global Model of Aerosol Processes (GLOMAP) scheme (Mann et al., 2010) with interannual variation derived from the EasyAerosol project (Stevens et al., 2017).

The MetUM atmosphere is coupled to the Joint UK Land Environment simulator (Jules, Best et al., 2011). Jules uses a nine-tile approach to represent sub-grid scale land cover heterogeneity, namely broadleaf and needle leaf trees, C3 and C4 grass, shrubs, inland water, bare soil, urban and land ice. Four soil levels are present with thicknesses of 0.1, 0.25, 0.65 and 2 metres. In BARPA-R, land surface properties are prescribed as per Walters et al. (2019), with the exception of the land sea mask, which is derived from the ERA Climate Change Initiative (CCI, Hartley et al., 2017), and the broadleaf canopy height, which is derived from Simard et al. (2011) following Dharssi et al. (2015). Land cover categorisation is fixed using a seasonal climatology following Hurtt et al. (2020).

## 2.2 Reference Datasets

This paper evaluates the performance of BARPA-R against three main observationally derived datasets: version 1 of the Australian Gridded Climate Dataset (AGCD; also known as AWAP), the ERA5 deterministic reanalysis, and the Australian Bureau of Meteorology's point-based station dataset. The current BARRA-V1 regional reanalysis is not used in this work as our core evaluation period goes back to 1985.

AGCD is a near-surface analysis product that uses an anomaly-based modified Barnes successive corrective method to interpolate gridded station data to a regular grid (Jones et al., 2009). In this work, AGCD version 1 is used to evaluate the ability of the BARPA-R system to reproduce the observed temperature and precipitation climate across Australian land points. The three AGCD variables used in this study, daily maximum temperature, daily minimum temperature and daily total precipitation,

are available on a regular grid with 0.05-degree latitude and longitude spacing. AGCD's performance hinges on the availability of station data, and so suffers from data availability issues in sparsely populated regions. A spatial mask, shown in Figure 2, is applied to precipitation metrics to remove the influence of regions most poorly constrained by observations, however observational uncertainty in the AGCD dataset remains.

Jones et al. (2009) describe key sources of observational uncertainty in AGCD. They highlight underestimations of maximum temperatures in regions of tight climate gradients and sparse observational coverage, including the coastal north-west Australia and the Nullarbor Plain due to poor resolution of maritime effects. They also note large analysis errors in daily precipitation estimates, with mean absolute errors up to 50% of the total. King et al. (2013) demonstrated that AGCD is suitable for the study of rainfall extremes, trends and variability across much of Australia, with limitations occurring in regions where station coverage is sparse. Meanwhile, Chubb et al. (2016) established large systematic dry biases between AGCD and an independent gauge network in the snowy mountains. In the following analysis, the direction of the AGCD biases is opposite to the BARPA-R bias presented. This means that the biases presented in this paper are likely overestimates, ensuring that our analysis is conservative.

ERA5 (Hersbach et al., 2020) is a global reanalysis product that combines data assimilation with ECMWF's Integrated Forecasting Sytem (IFS) model. As well as providing boundary conditions, ERA5 is used in the assessment of the BARPA-R evaluation simulation. In the performance evaluation section below, BARPA's biases are compared to ERA5's biases, both with respect to AGCD. However, since ERA5 benefits from assimilating observations while BARPA-R is a free running model within its regional boundaries, this reference is not regarded to be a minimum benchmark for some metrics. For example, it is not expected that BARPA-R will outperform ERA5 based on direct comparisons with observations at exact times and locations. When comparable levels of performance are present in BARPA-R and ERA5, this is interpreted a good result for BARPA-R. There are also expectations that some climatological metrics could indicate benefits from the BARPA-R downscaling, such as metrics based on spatio-temporal averages of weather conditions.

## 2.3 Evaluation Methodology

In section 3, the temperature and precipitation climatology is evaluated through analysis of derived standardised climate indices defined in the ICCLIM project (Pagé et al., 2022). These indices have been selected to evaluate aspects of the tails of the precipitation and temperature distributions, such as monthly maximum and minimum temperatures, and high precipitation rates. Indices have been computed on the 0.25° ERA5 grid following conservative remapping, aggregating from daily temperature extrema and precipitation data to monthly indices.

Performance was assessed over eight Australian regions, known as the National Resource Management (NRM) clusters (Clarke et al., 2015). These clusters are shown in Figure 1 and have been designed to be climatologically distinct and follow the boundaries of the Australia's 54 National Resource Management regions. This assessment is based on the decomposition of root mean square error into bias, correlation and variance error metric components following Su et al. (2013) and Gupta et al. (2009) and presented in equation 1. Error metrics selected were the seasonal biases, annual variance errors, climatological seasonal correlations and climatological spatial correlations. These error metrics are adjusted to reflect important climatological

aspects of model performance, as a like-for-like reproduction of observed weather events is not expected from free-running climate downscaling experiments.

$$\text{RMSE}^2 = \underbrace{\text{correl}(m,o)^2}_{\text{corr}} - \underbrace{\left(\text{correl}(m,o) - \frac{\text{stdev}(m)}{\text{stdev}(o)}\right)^2}_{\text{var error}} - \underbrace{\left(\frac{\frac{1}{n}\sum m - \frac{1}{n}\sum o}{\text{stdev}(o)}\right)^2}_{\text{bias}} \tag{1}$$

$$\text{bias}_{\text{DJF}} = \frac{1}{n}\sum_{x,y}\left(\frac{1}{90}\sum_{t\in\text{DJF}} m - \frac{1}{90}\sum_{t\in\text{DJF}} o\right) \tag{2}$$

$$\text{bias}_{\text{JJA}} = \frac{1}{n}\sum_{x,y}\left(\frac{1}{90}\sum_{t\in\text{JJA}} m - \frac{1}{90}\sum_{t\in\text{JJA}} o\right) \tag{3}$$

$$\text{var error} = \frac{1}{n}\sum_{x,y}\frac{\text{stdev}_{year}\left(\Sigma_{mon}\frac{m}{12}\right)+1}{\text{stdev}_{year}\left(\Sigma_{mon}\frac{o}{12}\right)+1} - 1 \tag{4}$$

$$\text{corr}_{\text{seas}} = \frac{1}{n}\sum_{x,y}\text{correl}_{mon}\left(\frac{1}{30}\sum_{year} m, \frac{1}{30}\sum_{year} o,\right) \tag{5}$$

$$\text{corr}_{\text{spatial}} = \text{correl}_{x,y}\left(\frac{1}{360}\sum_{time} m, \frac{1}{360}\sum_{time} o,\right) \tag{6}$$

Here, $m$ and $o$ represent the three-dimensional, monthly modelled and observed indices respectively, $\text{stdev}_x$ and $\text{correl}_x$ represent the act of computing the standard deviation or Pearson correlation of inputs over dimension $x$, and n indicates the number of gridpoints in the NRM cluster of consideration. The variance formula is modified by an offset of 1 to avoid division by zero in regions of low rainfall.

## 3 Performance Evaluation

### 3.1 Mean State

This section evaluates the performance of BARPA-R at simulating Australian monthly temperature and precipitation metrics as compared to AGCD and ERA5. Firstly, we examine the mean-state bias maps of seasonal-mean daily maximum and minimum temperatures and precipitation. Secondly, spatial and temporal characteristics of six temperature and four precipitation indices are examined, aggregated over the 8 NRM clusters. These indices were chosen with some emphasis on including properties of high impact weather. Finally, contemporary climate trends of the same ten indices are compared across the three data products.

Figure 2 displays seasonal bias maps over the Australian region of daily minimum temperature, daily maximum temperature and monthly precipitation totals, all averaged over the core evaluation period (1985-2014). Two seasons are presented here: December to February (DJF) and June to August (JJA). The remaining transition seasons are provided in Supplementary Figure A1. When temperature biases show a decrease in maximum temperatures coupled to an increase in minimum temperatures in the same season, this can be interpreted as an underestimation of the diurnal temperature range. During the Austral

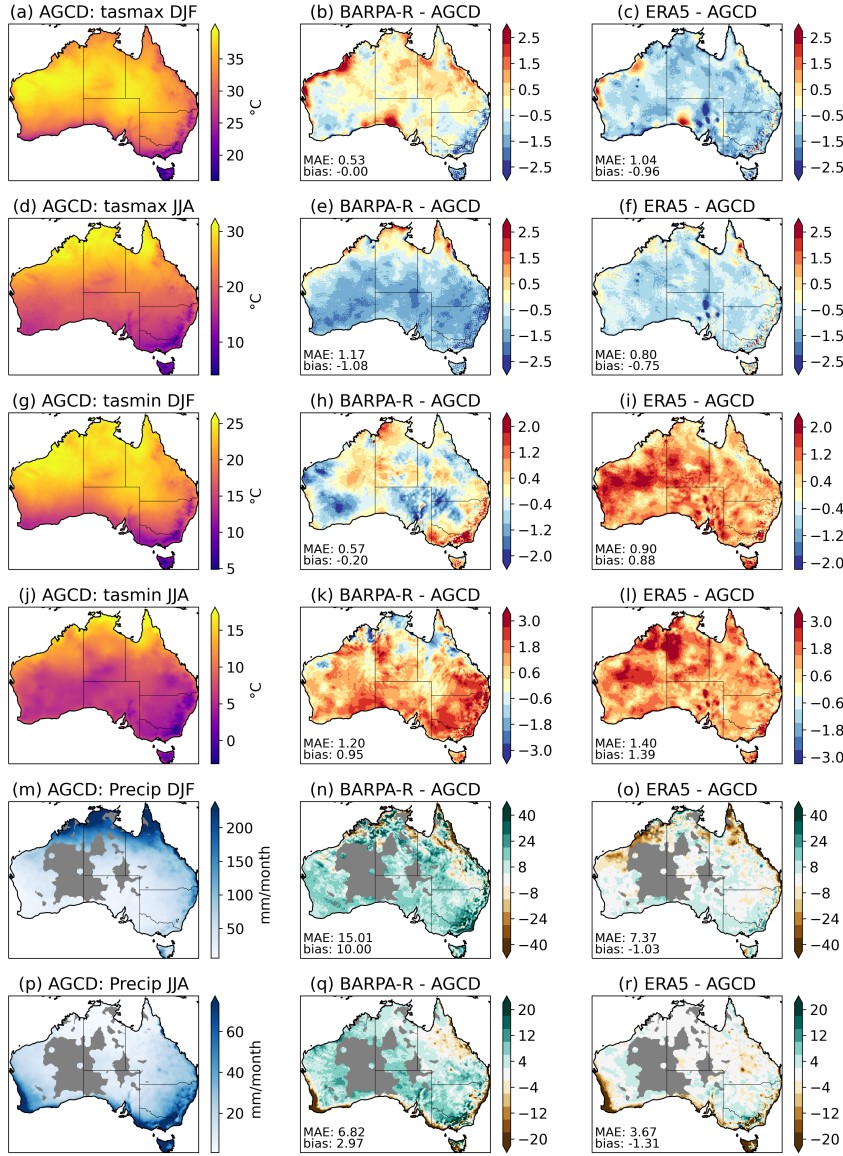

**Figure 2.** Bias in temperature and precipitation climate indicators (rows: TX, TN and PRCPTOT) for two seasons DJF and JJA, for BARPA-R and ERA5 (second and third columns) against AGCD (first column) averaged across the core evaluation period (1985-2014). The annotated figures indicate the area-averaged bias (top) and mean absolute error (bottom).

summer and northern Australian wet season, BARPA-R shows improvements in both daily minimum and daily maximum temperatures compared to ERA5, whose diurnal temperature range is reduced compared to observations across the country. However, BARPA-R does show a reduced diurnal temperature range across the south-east coast. Persistent warm biases of daily maximum temperatures in the Nullarbor may derive from observational uncertainty due to the low density of station data contributing to AGCD in these regions (Jones et al., 2009).

During the winter season, both BARPA-R and ERA5 show a reduced diurnal temperature range compared to the observed climate, with overly warm minimums and cool maximums, except for in the tropical north. The magnitude of the biases is higher in BARPA-R than ERA5, particularly on the highly populated East Coast. In all seasons, BARPA-R has a more realistic representation of Australia's inland lakes than ERA5.

The final two rows of Figure 2 show the monthly-aggregated precipitation biases. Overall, BARPA-R is overly wet, consistent with the overall performance of ACCESS-based models in the Australian region, including in NWP (Hudson et al., 2017). A prominent wet bias is present over the highlands in eastern Victoria in both seasons. However, wet biases surrounding the two masked regions in Western Australia (grey) are likely to be related to underestimates in AGCD due to the sparse station network (Jones et al., 2009). BARPA-R shows a reduction in ERA5's dry biases in southwestern Australia, western Tasmania, the Pilbara and Cape York.

Six temperature indices have been selected to examine BARPA-R's representation of Australia's regional temperature climates. The indices considered are: SU – number of summer days (Tmax $> 25$), TR – number of tropical nights (Tmin $> 0$), the monthly minimums and maximums of the daily minimums (TNn, TNx) and the same of the daily maximums (TXn, TXx). These indices have been computed on a monthly time-scale from daily maximum and minimum temperature data for AGCD and ERA5 and then regridded to the BARPA-R grid as described in section 2. Performance statistics described in section 2.3, namely biases, variance errors and correlations of the seasonal cycles, are calculated at each grid-point and then averaged across each NRM cluster. A spatial correlation was additionally calculated on the overall climatological mean of each index for each NRM cluster.

The resultant statistics are presented in Figure 3. The number of summer days is substantially improved in BARPA-R compared to ERA5, with reduced biases in most cases (save for summer in the north-most clusters), similar spatial correlations, and a much-improved seasonal cycle in the wet tropics. Tropical nights also show reduced biases but worse variance errors in many cases and worse performance in the South Slopes cluster. Absolute monthly maximum temperatures exhibit a strong cold bias in BARPA-R throughout the southern NRM clusters, consistent with the results shown in Figure 2.

The equivalent bar-charts for precipitation-based variables are shown in Figure 4. The metrics selected were number of RR1 – rain days (with at least 1mm of daily precipitation, R10m – heavy precipitation days (with at least 10 mm of daily precipitation), Rx1day – the monthly maximum daily precipitation amount, and SDII - the Simple Daily (precipitation) Intensity Index, which is calculated as the average precipitation rate across all days with at least 1 mm of precipitation. BARPA-R's wet bias is generally visible across the first three of these metrics, with BARPA-R biases generally tending towards more precipitation and rain days and being larger in magnitude than the ERA5 biases. Exceptions to this include the winter rain and heavy rain day count in the South and Southwest Flatlands, which are negative and reduced compared to ERA5, and rain days in the two

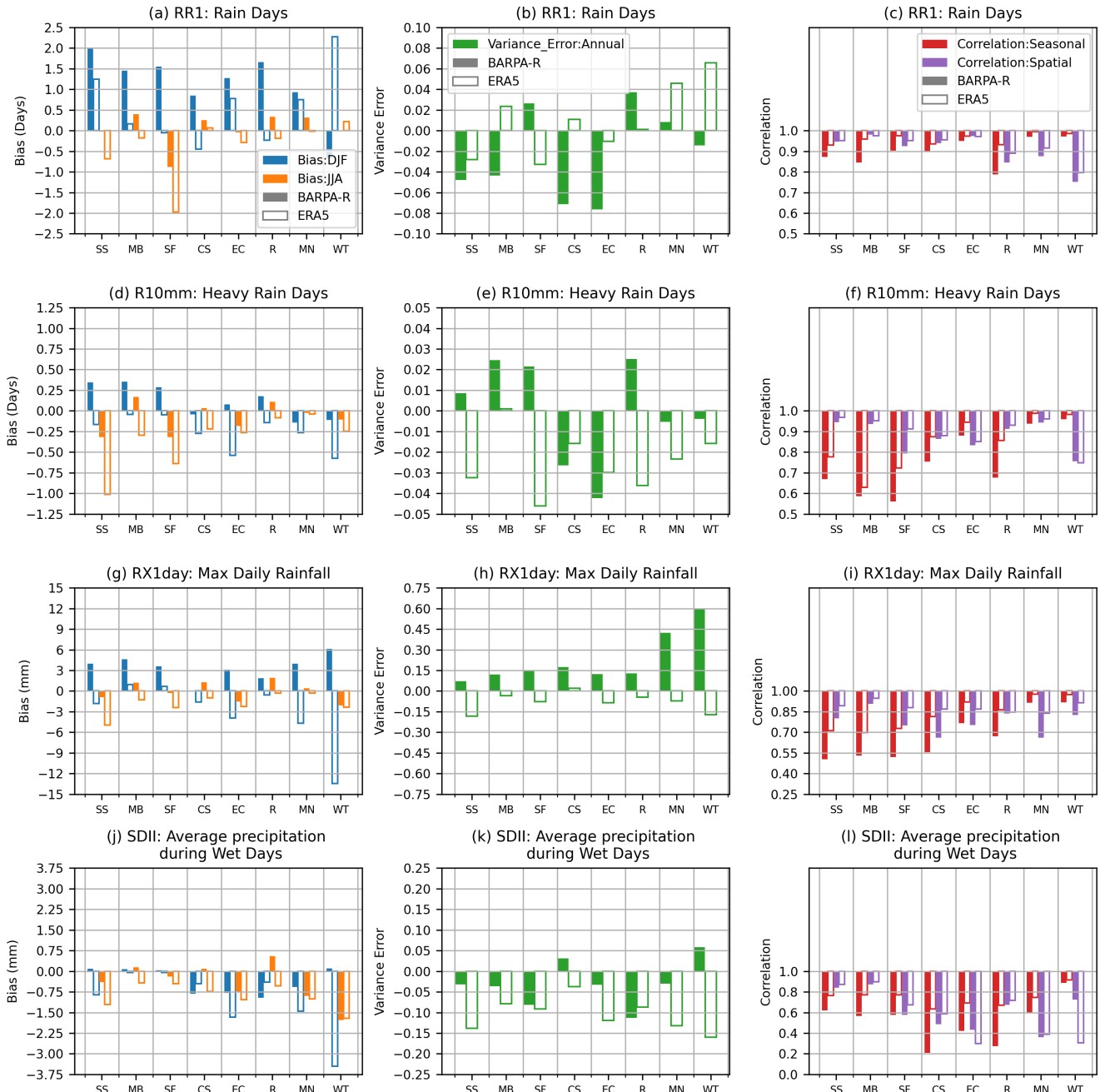

**Figure 3.** BARPA-R (solid bars) and ERA5 (outlined bars) performance of 6 temperature indices across the 8 Australian NRM clusters. Reference data is sourced from AGCD. Rows show number of summer days (SU; with daily maximum temperatures exceeding 25° C), tropical nights (TN; with daily minimum temperatures exceeding 20° C), and the monthly minimums and maximums of the daily minimums and maximums (TNn, TNx, TXn and TXx). Skill metrics are indicated by colour and column, with blue and orange showing the bias aggregated over summer and winter respectively (left), green representing the ratio of interannual standard deviations (middle), red representing the correlations in the climatological seasonal cycles and purple representing the spatial correlation across the NRM cluster of the climatological mean (right). All temporal metrics are computed at each grid-point and then spatially aggregated.

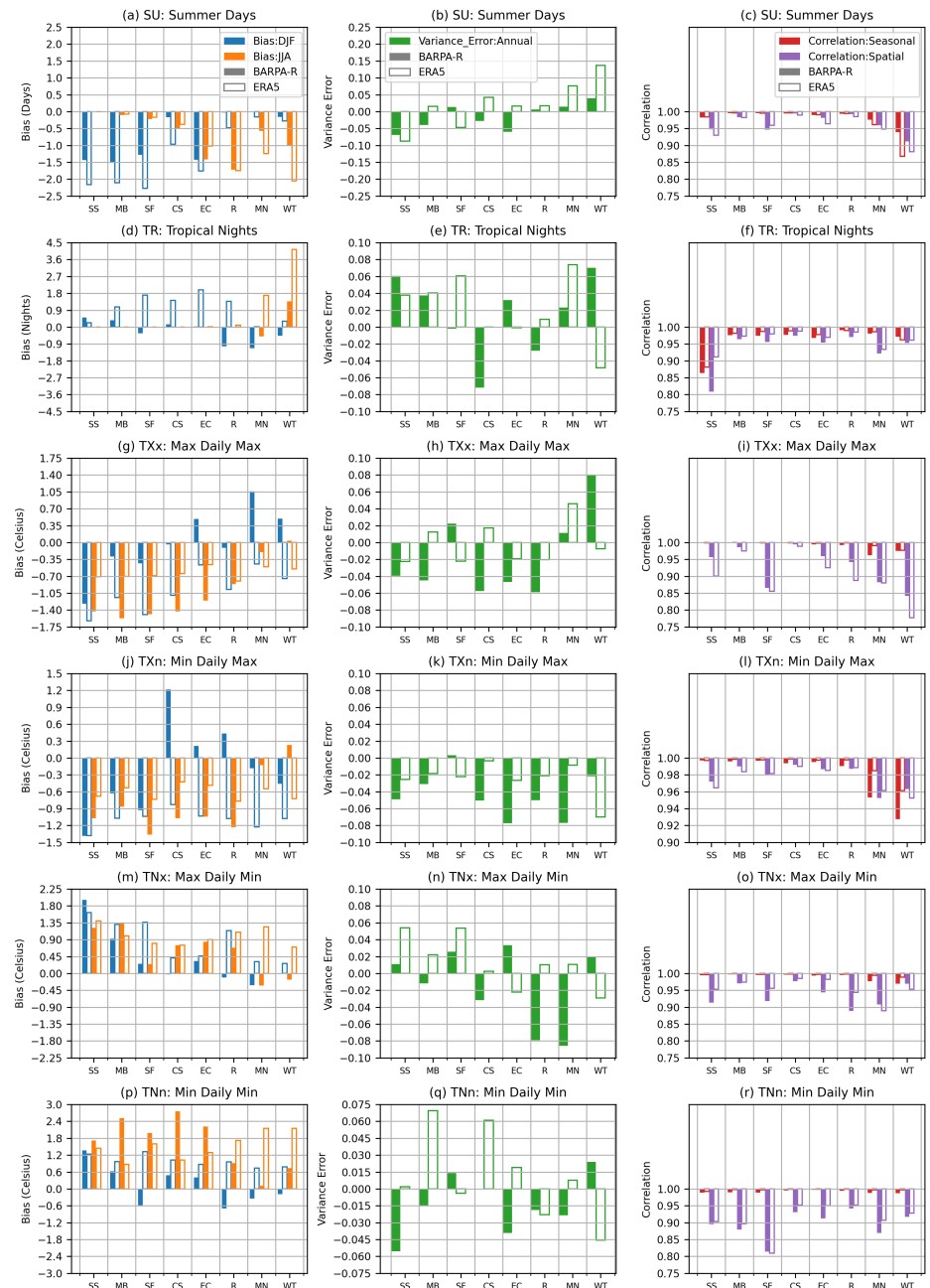

**Figure 4.** As per Figure 3 but for precipitation indices: wet days (RR1, > 1 mm/day), heavy rain days (R10mm; > 10 mm/day), monthly maximum daily precipitation (RX1day) and monthly precipitation (PRCPTOT).

tropical clusters where large positive biases in ERA5 are improved by BARPA-R. Maximum daily precipitation is consistently more variable on an interannual time-scale in BARPA-R than in AGCD across all NRM clusters. However, the SDII has a

consistent negative bias across ERA5 which is significantly improved in BARPA-R, particularly during the summer months. Both SDII and maximum daily precipitation have low spatial and seasonal correlation values, consistent with expectation that these fields will be quite noisy due to the influences of extreme values.

## 3.2 Trends

In order for the BARPA-R system to be of use in dynamically downscaling climate projections, it is crucial that BARPA-R is able to sensibly simulate changes in climate. Additionally, the subset of CMIP6 that will be downscaled with BARPA-R has been selected to cover a range of wetting/drying and high/low warming scenarios, with the intention that BARPA-R outputs can be used in a larger ensemble together with projections from other RCMs within CORDEX Australasia. Although it is possible that BARPA-R may be found to diverge from it's host GCM for good reasons, this model selection was based on the hypothesis that this spread in future change will be translated to some degree into the BARPA-R ensemble. Therefore, this section investigates the degree to which BARPA-R is able to simulate observed trends in contemporary climate.

The study periods used for this analysis are two 10-year time-slices: 1985–1994 and 2005–2014. Due to their short durations, these time-slices will include a degree of interannual variability as well as any anthropogenic climate change. However, it is expected that this variability will be in phase and consistent across the observations and driving reanalysis data, and therefore should be reproducible by BARPA-R.

This trend analysis must be caveated by the observational uncertainties associated with the trends of both AGCD and ERA5. Long-term trends in observational datasets, including analyses and reanalyses are sensitive to temporal inhomogeneities in their input datasets (Gibson et al., 2019, e.g.). Simmons et al. (2021) found that temperature trends over Australia are affected by inhomogeneities in the observational inputs, however the poorest performance occurs prior to 1970, before our study period. AGCD has been designed to be more robust to long-term trends, through the application of an anomaly-based approach which takes advantage of climate normals at a subset of stations with longer coverage. Jones et al. (2009) demonstrate that this approach provides consistent maps of precipitation trends compared to monthly analyses derived only from stations with long climate records and found that temperature trends were similarly robust at the large scale.

With these caveats in mind, this paper accounts for observational uncertainty in rainfall trends by focussing attention on established trends that have been studied elsewhere, namely southern Australian cool season drying, wetting trends in north-western Australian during summer, and the intensification of short-duration heavy convective rainfall (Tolhurst et al., 2023; Borowiak et al., 2023; Fowler et al., 2021).

The contemporary change the temperature-based ICCLIM indices across BARPA-R, AGCD and ERA5 are shown in Figure 5. BARPA-R shows warming trends for all the indicators, across all the clusters. There is some consistency between BARPA-R and AGCD, particularly for the indicators based on maximum temperatures. Statistically significant AGCD trends present in montly mean maximum temperature (TX), summer days (SU) and monthly maximum temperature (TXx) in the southern clusters are generally well captured and significant in both BARPA-R and AGCD. However, minimum temperature-based indices show increased rates of warming in BARPA-R that are not reflected in the observed products. Aside from in the Murray Basin cluster, these changes are not statistically significant at the $p < 0.05$ level. However, some cooling trends are observed

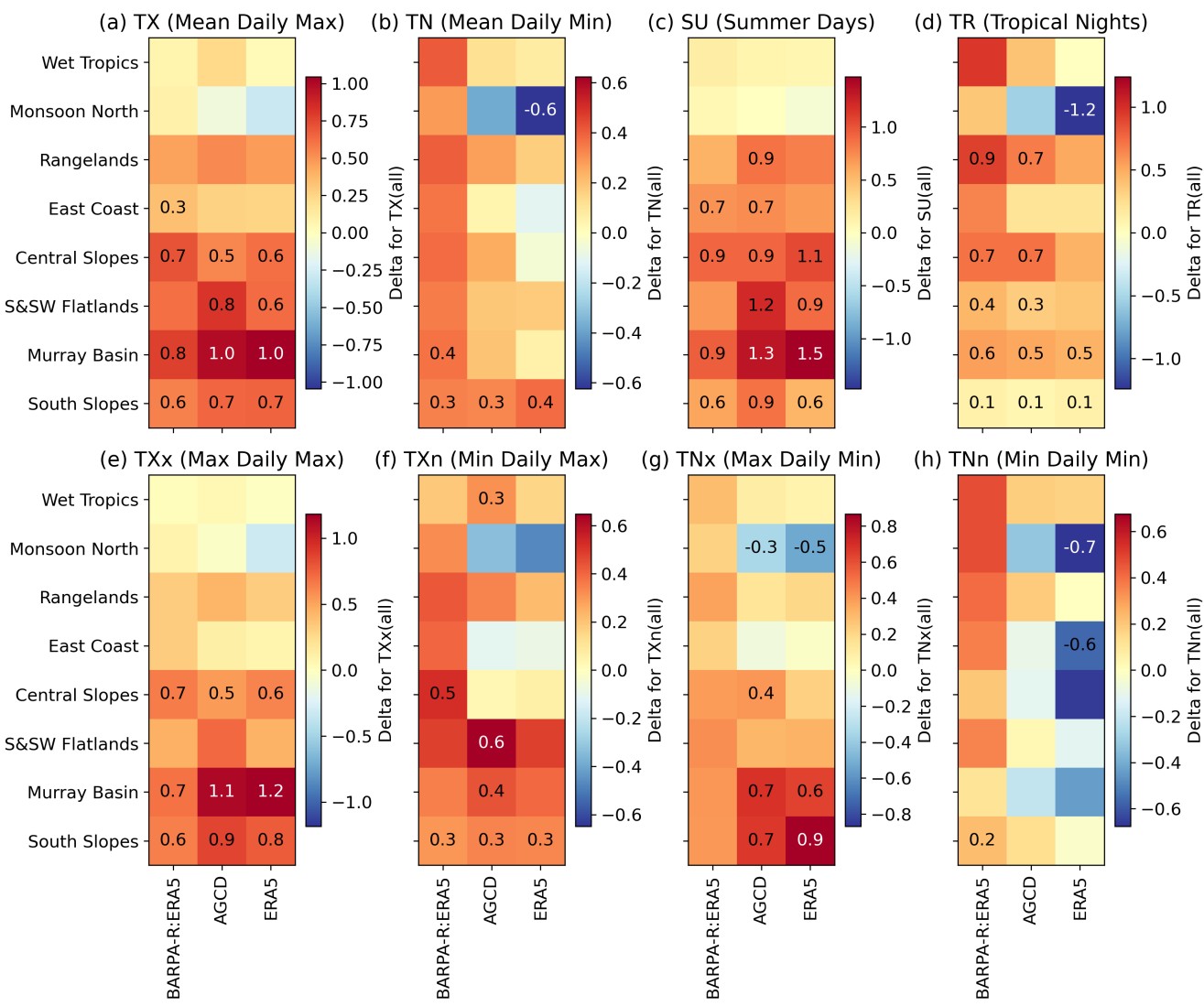

**Figure 5.** Contemporary change in annual means of 8 temperature indices between the period 1985-1994 and 2005-2014 aggregated across NRM clusters. Indices are as per Figure 3, together with the monthly mean daily temperatures (TX and TN). Values are annotated on the figures when the early and late samples are significantly distinct at the $p > 0.05$ level using a Welch's t-test.

in AGCD and ERA5 that are not present in BARPA-R, most noticeably in the Monsoonal North and in absolute minimum temperatures.

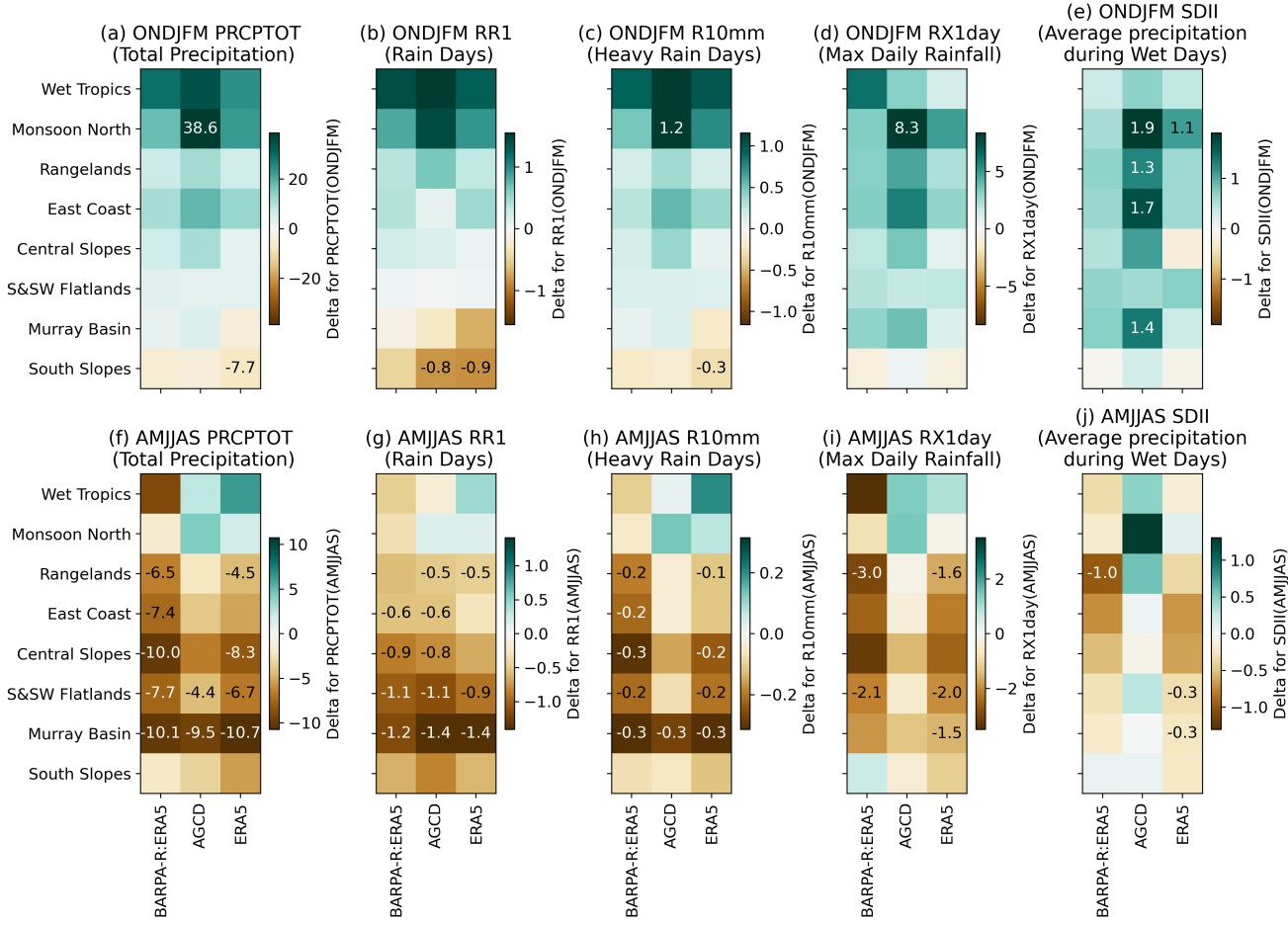

**Figure 6.** Contemporary change in seasonal means of 5 precipitation indices between 1985-1995 and 2005-2015. Indices as per Figure 4, together with monthly total precipitation (PRCPTOT). Values are annotated on the figures when the early and late samples are significantly distinct at the $p > 0.05$ level using a Welch's t-test.

Corresponding trend plots for precipitation indices are shown in Figure 6. As contemporary trends show a strong seasonal dependence, these trends have been split into warm season (October – March) and cool season (April – October) panels. The direction of change is generally consistent across all three datasets in the warm season. Significant AGCD-based increases in precipitation intensities across multiple NRM clusters (Figure 6e), are not reflected in either BARPA-R or ERA5. This result highlights the difficulty that parameterised convection models and reanalysis products have in simulating the observed intensification of short duration extreme precipitation (Fowler et al., 2021). A decrease in the number of dry days in the Southern Slopes region is evident in ERA5 but insignificant in BARPA-R. Conversely in the cool season, drying trends are

typically more pronounced in BARPA-R and ERA5 than in AGCD. Reductions in rain days are consistent across all three datasets.

## 3.3 Interannual Variability

BARPA-R outputs are examined here in relation to three key modes of interannual climate variability: the El Niño-Southern Oscillation (ENSO), the Indian Ocean Dipole (IOD) and the Southern Annular Mode (SAM). These modes of variability typically have the largest observed teleconnections to Australian climate during the Austral Spring (September to November), so this section focuses on that season. In order to increase the sample size of modes of variability, the full 42-year period from 1979 to 2020 has been sampled.

Figure 7 shows the composite differences between the active phases of each mode of variability and the climatological means for precipitation and daily maximum temperatures, aggregated across the NRM clusters. Precipitation anomalies are presented as percentages of the climatological mean. Spatial variability in the IOD teleconnection is very similar across all three datasets. In the northern clusters, precipitation anomalies during the positive phase of the SAM are too weak in BARPA-R and do not reflect AGCD's statistical significance. BARPA-R also misses significant warm and cool temperature anomalies in the Central Slopes and East Coast clusters due to both phases of ENSO. However, there is a remarkably close correspondence between maximum temperature and the SAM across BARPA-R and AGCD. Overall, all three teleconnections are well represented by BARPA-R.

## 3.4 10-metre Winds

In the absence of a gridded wind analysis, near-surface wind speeds have been evaluated against 3-hourly station observations taken from 10-metre masts. Where quality information was present, observations that were flagged as wrong, suspect or inconsistent were excluded from the analysis. Model and reanalysis data corresponding to the observations were extracted from ERA5 and BARPA-R. For each station, instantaneous wind speed data for a height of 10m AGL was extracted from the nearest grid-cell to the station position. The model dynamical time-steps (7.5 minutes for BARPA-R and 12 minutes for ERA5) roughly correspond to the observational averaging period (10 minutes), which ensures that the modelled and observed wind speeds are comparable. Only time samples for which valid station data was present are considered. The resulting model and observation data were then aggregated to NRM cluster level.

Resulting quantile-quantile (Q-Q) plots of observed and corresponding modelled 10-metre wind speed for each NRM cluster are presented in Figure 8. The Perkins Skill Score (PSS Perkins et al., 2007) has been used to compare the distributions of BARPA-R and ERA5 to the observed station wind speeds, and is listed in the captions of Figure 8. The PSS measures the difference between two normalised distributions, ranging between 1 for a perfect match to 0 for no overlap between distributions, and is sensitive to histogram bin width, in this case 0.5 m/s. In six of the eight NRM clusters, BARPA-R shows an improved PSS and improved 99th percentile wind speeds compared to ERA5. BARPA-R generally shows improved high percentile tail values compared to ERA5, while both models underestimate 'calm' weather conditions with wind speeds of 0 m/s. In the upper tail, there is a general tendency for both BARPA-R and ERA5 to have the Q-Q line tending towards lower

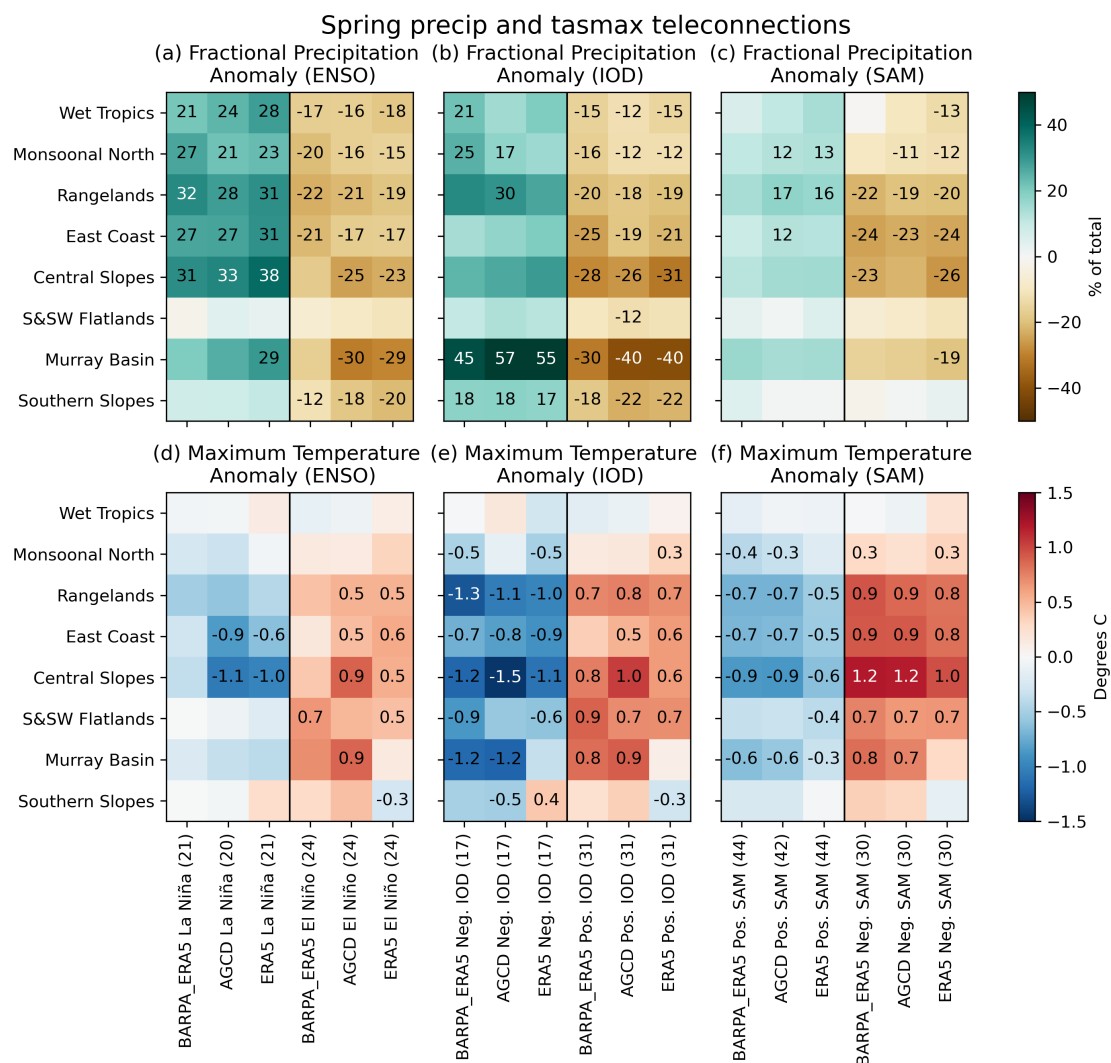

**Figure 7.** Spring fractional precipitation (top) and maximum temperature (bottom) composite anomalies under positive and negative phases of (left): ENSO, (centre): IOD and (right): SAM. Values are annotated on the figures when the composite anomaly is statistically significant from 0 at the $p > 0.05$ level using a Welch's t-test.

values, similar to previous results for BARPA-R downscaling of CMIP5 simulations which also found an improvement for this when using BARPA-C convection-permitting simulations (Dowdy et al., 2021). This is as expected to some degree given the very strong winds from some localised storms may be better simulated at finer scales.

In summary, performance evaluation of precipitation and surface air temperatures has demonstrated that BARPA-R is capable of producing a faithful representation of present-day climate when deriving driving inputs from ERA5. BARPA-R shows a persistent wet bias across a set of precipitation-related indices, and a winter cold bias in maximum temperatures. Maximum temperature trends are broadly consistent with observations, while warming trends in minimum temperatures are over-estimated. Precipitation trends resemble ERA5 more closely than AGCD, and while the cool-season drying in southern Australia is well captured, deficiencies in simulating the intensification of heavy precipitation by parameterised convection models is evident in both BARPA-R and ERA5. Regional correlations with key modes of variability, namely ENSO, IOD and SAM, are well simulated. 10-metre winds are improved over ERA5 but still under-estimate the high-tails of the distribution in many regions.

## 4    Process Evaluation

This section provides an analysis of the BARPA-R's representation of some key atmospheric dynamical and thermodynamical processes that are important for the Australian Region. Focus is placed on key wind circulation features and on large-scale weather systems. Firstly, the climatologies of these features are compared between BARPA-R and observational datasets. This climatological analysis is provided to demonstrate the fidelity with which BARPA-R reproduces regional climate process. Secondly, interannual correlations of location and frequency statistics for each circulation feature or weather system are computed between BARPA-R and the real-world observations. This correlation analysis demonstrates the degree to which the weather and circulation is coupled with the boundary conditions, versus the degree to which these systems are free to evolve independently within the model.

### 4.1    Circulation

Figure 9 shows heatmaps of the frequency of the presence of three key large-scale circulation features of the Australian region across four seasons: the barotropic and subtropical jets, and the monsoonal westerly winds. Table 1 further shows the biases and interannual correlations with ERA5 key properties of each circulation feature. In this analysis, ERA5 is used as the reference dataset. The computational methods apply simple thresholds to daily mean wind speeds to determine the horizontal locations of each circulation feature. The occurrence frequencies are likely to be somewhat sensitive to the choice of thresholds, however, further analysis (not shown) has found that BARPA-R model biases are robust to threshold choice. The location of the South Pacific Convergence Zone is also shown. Feature definitions are given below:

– Barotropic Westerly Jet (blue): 850 hPa and 200 hPa zonal winds both exceed 10 m/s.

– Monsoon Westerlies (green): 850 hPa zonal wind is westerly, while 200 hPa zonal wind is easterly.

– Subtropical Jet (red): 200 hPa zonal winds exceed 30 m/s.

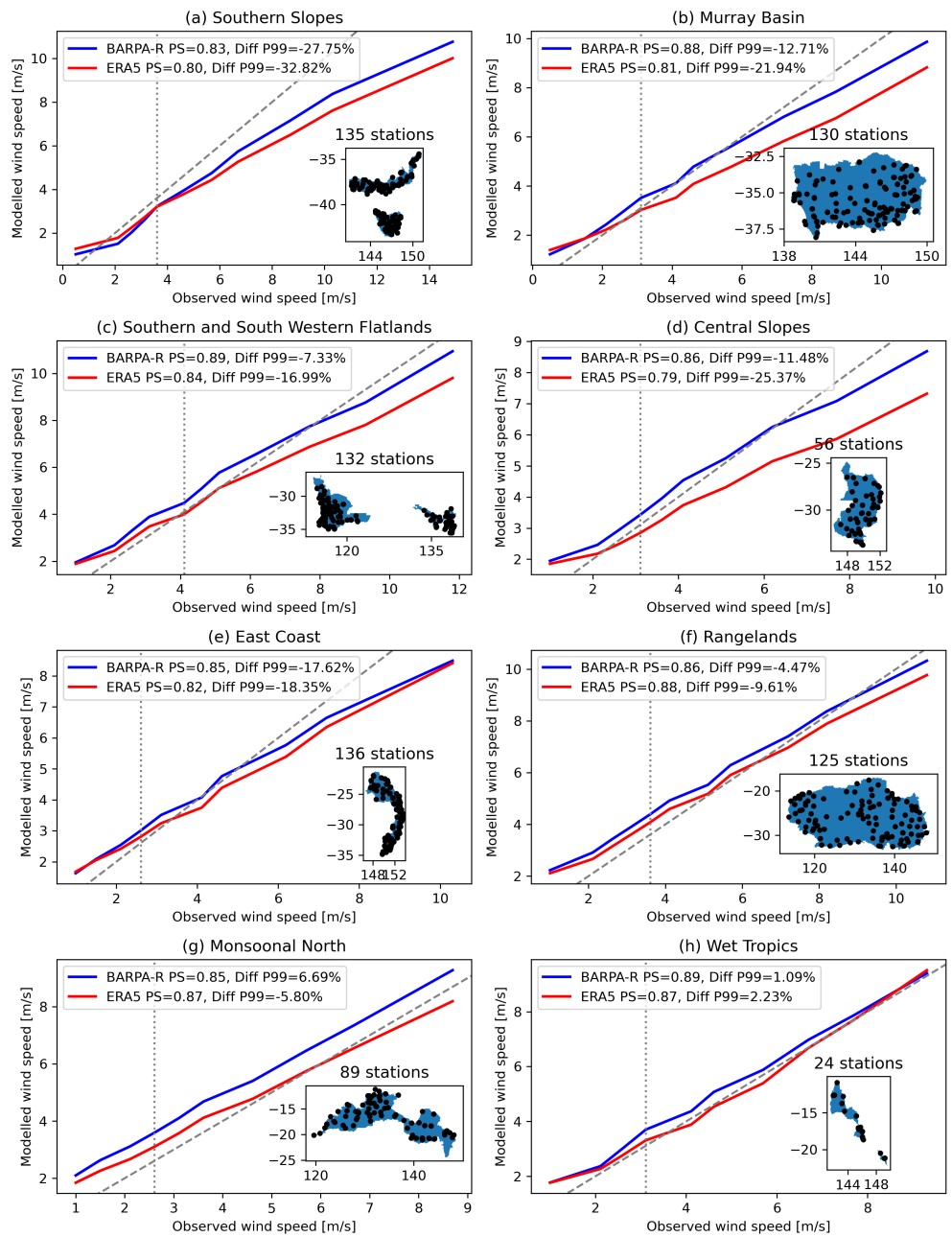

**Figure 8.** Q-Q plots of observed and modelled hourly wind speeds at station locations in each NRM clusters of ERA5 (red) and BARPA-R (blue), compared to station observations. Perkins Skill Scores and 99th percentile biases are given in each figure legends. Station locations are shown as black dots in the inset maps. Model data is derived from time-step instantaneous winds and interpolated to station locations using a nearest neighbour interpolation scheme. The number of stations and hours are given in each figure label.

- South Pacific Convergence Zone (SPCZ, orange): Linear fit to the latitude of the monthly maximum of precipitation in
the South-West Pacific, for each longitude between 150 and 200 E. This methodology is modified from Brown et al.
(2013) for the BARPA-R domain. In Figure 9, the orange marker indicates the interannual inter-quartile range of the
seasonal SPCZ location.

All four features are present in both BARPA-R and ERA5 in Figure 9, with matching seasonal cycles. Some biases are
evident, however, which are further summarised in Table 1. The largest biases are present in the monsoon westerlies, which
are shifted too far east, particularly during the boreal monsoon, and the SPCZ, which is shifted south in March to May. The
spatial extent of the Subtropical Jet is additionally reduced in all seasons. The bias in the monsoon westerlies is a persistent
systematic MetUM bias (Rodríguez and Milton, 2019). This bias has been linked by **?** to errors in the representation of
convection over the Maritime Continent and the western-central equatorial Indian Ocean. Systematic rainfall biases in the
maritime continent are common due to the complex, multi-scale nature of convection in this region. A reduced southerly bias
in the SPCZ location has been documented in the ACCESS-S1 seasonal forecast system (**?**), suggesting that ocean coupling
may improve the representation of the SPCZ.

The right-hand side of Table 1 shows the correlations between the circulation system indices (latitude, longitude, spatial
extent and SPCZ slope) in ERA5 and BARPA-R. These correlations are not measures of model performance, as it is not
required that BARPA-R shows perfect agreement in interannual variability phasing as its driving model. Instead, they show
where circulation systems are influenced by the internal variability of the BARPA-R system, and where they are constrained by
boundary conditions and SST forcing. From the table, it is evident that tropical features, namely the SPCZ and the Monsoon
Westerlies, have a larger degree of internal variability, while the subtropical and barotropic jets are more constrained and remain
in phase with ERA5.

**Table 1.** Bias and interannual correlations of circulation features compared to ERA5.

| Feature | Index | Units | Biases | | | | Correlations | | | |
|---|---|---|---|---|---|---|---|---|---|---|
| | | | DJF | MAM | JJA | SON | DJF | MAM | JJA | SON |
| Subtropical Jet | Latitude | Deg Lat | -0.17 | -0.02 | 0.01 | -0.10 | 1.00 | 1.00 | 1.00 | 1.00 |
| | Longitude | Deg Lon | -0.31 | -0.15 | -0.09 | -0.14 | 0.99 | 0.99 | 0.99 | 1.00 |
| | Extent | % grid | -0.53 | -0.52 | -0.37 | -0.47 | 0.99 | 0.98 | 0.99 | 0.99 |
| Monsoon Westerlies | Latitude | Deg Lat | -0.23 | 0.20 | 0.47 | 0.40 | 0.81 | 0.81 | 0.72 | 0.87 |
| | Longitude | Deg Lon | 0.78 | 1.37 | 3.19 | 1.94 | 0.96 | 0.93 | 0.93 | 0.98 |
| | Extent | % grid | -0.23 | -0.33 | 0.25 | 0.62 | 0.9 | 0.92 | 0.97 | 0.97 |
| Barotropic Jet | Latitude | Deg Lat | 0.08 | -0.11 | -0.17 | -0.05 | 0.91 | 0.95 | 0.98 | 0.98 |
| | Longitude | Deg Lon | 0.02 | -0.03 | -0.03 | -0.08 | 0.97 | 0.99 | 0.98 | 0.99 |
| | Extent | % grid | 0.00 | -0.04 | -0.09 | -0.01 | 0.98 | 0.98 | 0.99 | 0.99 |
| SPCZ | Latitude | Deg Lat | -1.34 | -2.73 | -0.88 | -0.17 | 0.74 | 0.84 | 0.76 | 0.81 |
| | Slope | 1 | -0.05 | -0.04 | 0.00 | -0.01 | 0.19 | 0.53 | 0.48 | 0.71 |

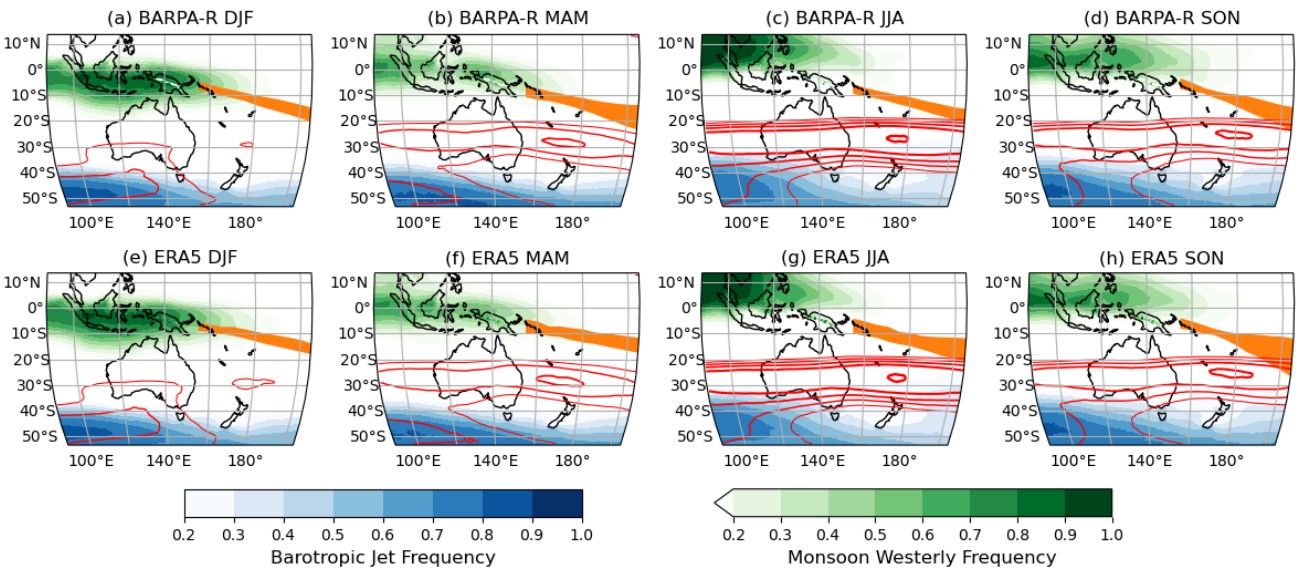

**Figure 9.** Heatmaps of seasonal circulation feature fractional frequency, ranging from 0 to 1, in BARPA-R (top) and ERA5 (bottom). Colours indicate: the westerly jet at 850 hPa (blue), the monsoonal westerlies at 850 hPa (green), the subtropical jet at 200 hPa (red lines; contour interval = 0.15, first contour: 0.2). Additionally, the location of the SPCZ is shown in orange. Feature definitions are provided in the text.

## 4.2 Weather Systems

Figure 10 and Table 2 follow the format of Figure 9 and Table 1, but consider a set of large-scale weather systems that influence Australia, namely tropical and extra-tropical cyclones, and Australian Northwest Cloud-Bands (NWCBs). Where weather features only occur in limited seasons are not necessarily observed in every year, interannual correlations are only given for the feature counts and statistics are only shown in seasons when the weather systems are present. In this analysis, some direct observational products are available, and these are used as references where possible. Where no direct observation

is available, ERA5 is used as the reference. Identification algorithms and reference datasets are described below.

Firstly, tropical cyclones are identified using the Okubo-Weiss-Zeta (OWZ) methodology following the methodology of Tory et al. (2013) and Bell et al. (2018). This algorithm uses a low-deformation vorticity parameter derived from vorticity and deformation parameters at 850 and 500 hPa, and tropical cyclone environment parameters derived from relative and specific humidity at 950 and 700 hPa. The reference dataset is the International Best Track Archive for Climate Stewardship (IBTrACS

Knapp et al., 2010). In Table 2, tropical cyclones are split into eastern and western systems along the longitude band at 135° E, corresponding to the Indian Ocean and West Pacific Ocean tropical cyclone basins. Secondly, extra-tropical cyclones are identified using the University of Melbourne (UM) tracker (Pepler and Dowdy, 2021) by identifying local minima in mean sea level pressure for which the maximum sea-level pressure Laplacian exceeds 0.8 hPa/deg lat$^2$ and which originate south of 35°S. In this case, tracks derived with the same algorithm using ERA5 reanalysis are used as the reference dataset.

Finally, NWCBs are identified using the MetBot (Hart et al., 2012). This algorithm identifies bands of continuous low daily Outgoing Longwave Radiation (OLR) spanning from the tropics through the subtropics, and has been used to identify similar weather systems in Southern Africa and South America. In this Australian application, the OLR threshold has been set to 240 K in observations, and 255 K in BARPA-R, with the latter selected through matching quantiles of daily OLR. Each NWCB must intersect the longitude range, 110°–155° E along each latitude band between 29° and 11°S.

Together, Figure 10 and Table 2 show that extra-tropical cyclones are well represented in BARPA-R across all seasons. There is a westward bias in feature locations, and high correlations above 0.8 across BARPA-R and the ERA5-based reference. Tropical cyclones are generally shifted south and west, and the large spike in cyclone systems in north-western Australia is underestimated. Tropical cyclone interannual variability is decoupled from observations, with very low and even negative correlation values present. Further investigation (not shown) indicates that tropical cyclone locations and paths diverge on 390 seasonal and sub-seasonal timescales between BARPA-R and observations away from the domain boundaries. Finally, the spatial distribution of NWCBs has the correct shape, with a maximum over the Australian East Coast in the DJF season. However, cloud-band counts are reduced by 13% in this core NWCB season. Interannual correlations with observations are 0.5 and 0.66 in DJF and MAM respectively, suggesting a degree of coupling with the boundary conditions as well as real-world interannual variability.

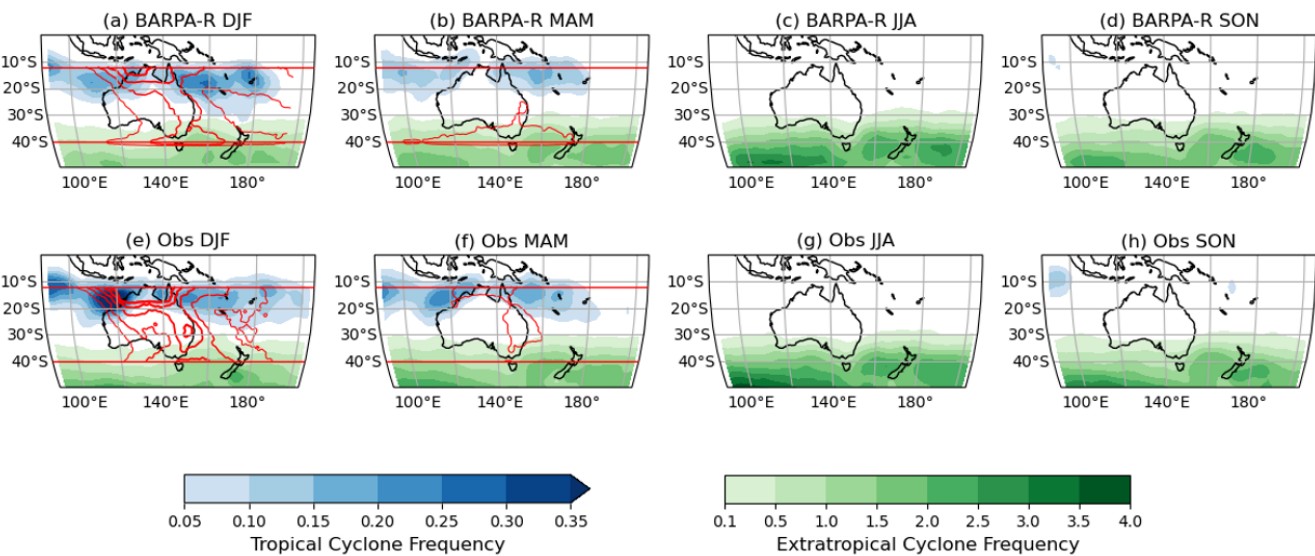

**Figure 10.** Heatmaps of seasonal weather feature frequency, in units of events per square degree per season, in BARPA-R (top) and observations (bottom). Colours indicate: tropical cyclones (blue), extra-tropical cyclones (green) and NWCBs (red lines; contour interval: 1 event/season, starting value: 2). Feature definitions are provided in the text. Observational products vary by feature: IBTRaCS tropical cyclones, ERA5 extra-tropical cyclones, and NOAA satellite-derived daily OLR-based cloud bands.

**Table 2.** Bias and interannual correlations of weather features compared to IBTRaCS, ERA5 and NOAA OLR as per text.

| Feature | Index | units | Metric | DJF | MAM | JJA | SON |
|---|---|---|---|---|---|---|---|
| Tropical Cyclone (East) | Latitude | Deg Lat | Bias | -2.27 | -0.41 | - | - |
| | Longitude | Deg Lon | Bias | -2.10 | -2.71 | - | - |
| | Count | % diff | Bias | 5.5 | -2.5 | - | - |
| | Count | 1 | Correl | 0.30 | 0.09 | - | - |
| Tropical Cyclone (West) | Latitude | Deg Lat | Bias | -0.77 | -0.91 | - | - |
| | Longitude | Deg Lon | Bias | -0.79 | -2.50 | - | - |
| | Count | % diff | Bias | -29.1 | -30.1 | - | - |
| | Count | 1 | Correl | -0.07 | 0.17 | - | - |
| Extratropical Cyclone | Latitude | Deg Lat | Bias | 0.23 | 0.10 | 0.03 | -0.01 |
| | Longitude | Deg Lon | Bias | 1.08 | 1.60 | 1.63 | 2.00 |
| | Count | % diff | Bias | 0.13 | -1.90 | 0.85 | 1.09 |
| | Count | 1 | Correl | 0.83 | 0.87 | 0.92 | 0.84 |
| Northwest Cloud-Band | Latitude | Deg Lat | Bias | 2.47 | 2.58 | - | - |
| | Longitude | Deg Lon | Bias | -1.09 | -2.08 | - | - |
| | Count*check | % diff | Bias | -13.6 | 9.7 | - | - |
| | Count*check | 1 | Correl | 0.51 | 0.66 | - | - |

## 5 Lagged Temperature-Precipitation Relationship

Correct simulation of multivariate relationships between RCM output variables are important for accurately representing weather processes, compound events and downstream impact modelling, which take multiple inputs from RCMs (Kim et al., 2021, 2023; Sain et al., 2011). Therefore, it is important to assess how well BARPA-R captures multivariate relationships, in particular, between key variables like temperature and precipitation, as compared to existing observational and reanalysis datasets.

A useful metric for characterising the relationship between two variables is their time-lagged correlation, which can indicate how each variable responds to anomalies of the other through examination of positive and negative lags respectively. Hence, the lagged correlations between these variables may be useful to examine the time lag and determine the strength and direction of the relationship between them (Kumar et al., 2013). At longer timescales, lagged correlations can also be helpful to identify potential feedback mechanisms between precipitation and temperature. For instance, if increased precipitation leads to cooler temperatures, this can lead to enhanced vegetation growth, which can further increase precipitation due to amplified transpiration and evaporation. In convective climates, positive correlations at negative lags may be linked with atmospheric instability as the land heats up, thus, making conditions favourable for convection to occur; while negative correlations at positive lags suggest that the precipitation cools the surface due to evaporation and cloud cover, resulting in lower temperatures. Moreover, positive correlations at positive lags (especially, in the minimum temperatures) may be associated with increased cloudiness

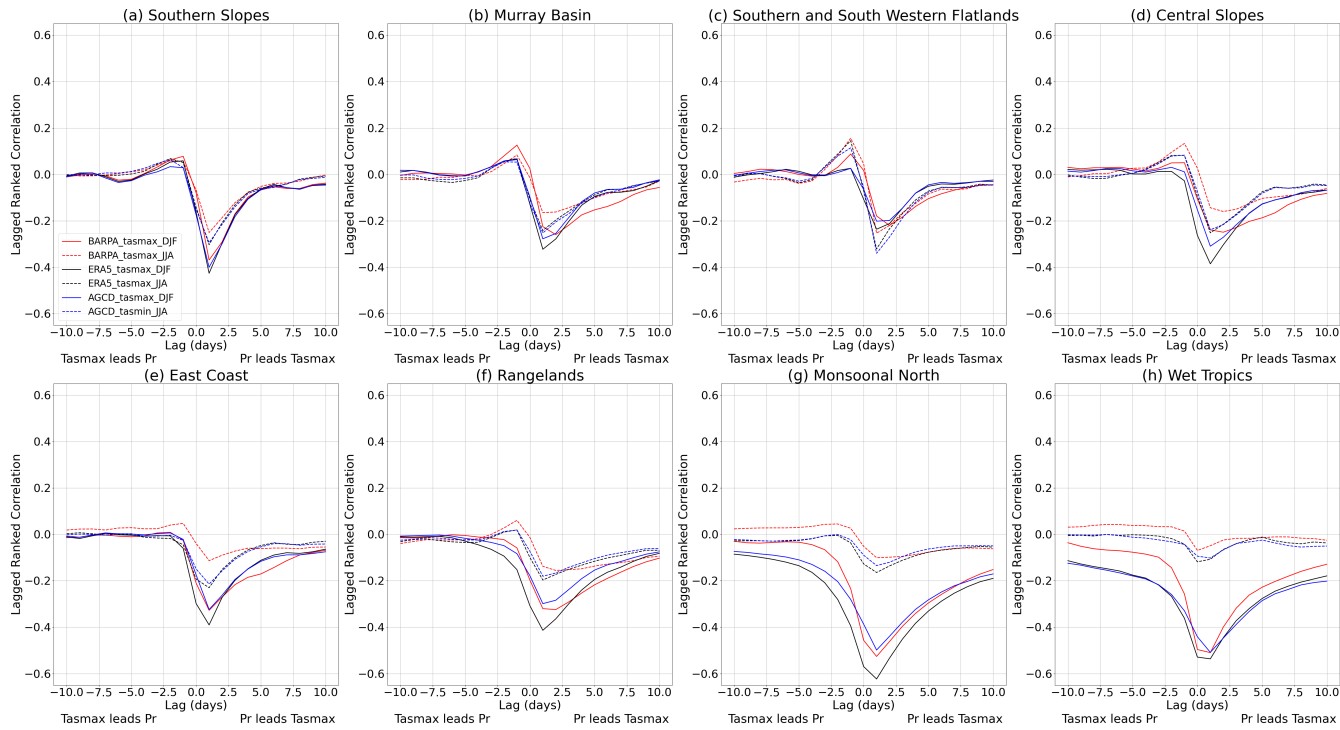

**Figure 11.** Lagged Spearman ranked correlations between daily precipitation and maximum temperature (tasmax).Lines indicate BARPA-ERA5 (red), ERA5 (black) and AGCD (marked in blue) in DJF (solid lines) and JJA (dashed lines) over the eight NRM clusters across Australia (as labelled). Daily AGCD and modelled precipitation data are set to zero where the values are less than (<) 1mm/day. The correlation is computed at each grid point, before spatially averaged over each region.

thereby increasing the chances of instability and precipitation; additionally, an increase in warm and humid conditions is expected, leading to higher temperatures.

This section evaluates the daily temperature-precipitation relationship in BARPA-R and compared to AGCD and ERA5. Seasonal Spearman's ranked correlations with lag time of $\pm 10$ days are computed between the daily maximum/minimum temperature and precipitation outputs from 1985 to 2014. A lower precipitation threshold of 1mm/day was applied before ranking the precipitation data to remove sensitivity to data storage precision. incorporating a precipitation threshold of 1 mm/day, assuming that this is the minimum amount of precipitation required to be considered as a precipitation event for a particular day. The timesteps of AGCD maximum temperature data were shifted by 1 day to ensure that valid times were consistent across all datasets.

Figures 11 and 12 show the lagged Spearman ranked correlations between daily precipitation and near-surface minimum/-maximum temperatures (tasmin/tasmax) in the different datasets, namely, BARPA-R, ERA5 and AGCD in DJF and JJA over the eight NRM clusters. The remaining seasons showed similar results (not shown). The lagged temperature-precipitation correlation relationships between ERA5 and AGCD are very similar across seasons and NRM clusters.

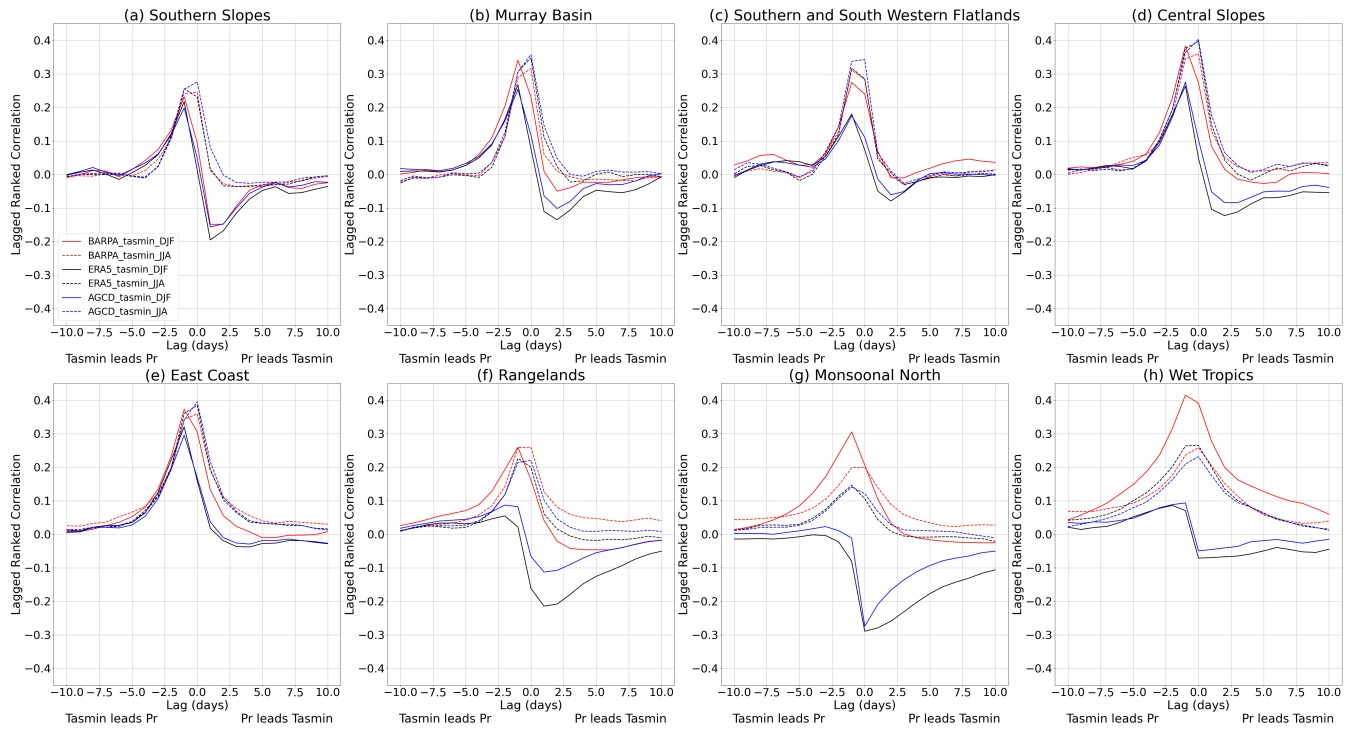

**Figure 12.** Lagged Spearman ranked correlations between daily precipitation and minimum temperature (tasmin). Lines and subplots as per Figure 11.

In the mid-latitude regions (Figure 11 a-e), precipitation generally leads tasmax with negative correlation at positive lag of around 1 day, suggesting that precipitation initially cools the surface, leading to lower maximum temperature. This is consistent across all three datasets in all seasons. In all three datasets and in both seasons, tasmin leads precipitation (Figure 12 a-e) with positive correlation at negative lag of around 1 day (which may accelerate evaporation, leading to an increase in atmospheric moisture and condensation). Seasonal differences in the tasmin-precipitation relationship are well distinguished by BARPA-R in the Southern slopes, Murray Basin and SSW Flatlands regions, while in the Central Slopes (Figure 12 d) and East Coast (Figure 12 e) the BARPA-R DJF relationships more closely resembles the observed relationships in JJA.

In north-central Australia, the observed precip-tasmin relationship is distinctly different between DJF and JJA (Figure 12 f-h). In JJA, this relationship is characterised by positive correlations and is well simulated by BARPA-R. However, in DJF, negative correlations are seen at a positive lag of around 1 day in both AGCD and ERA5. However, BARPA-R still shows positive correlations at negative lag (Figure 12 f-h), resembling its relationship in JJA. In these regions, namely, the Rangelands, Monsoonal North, and Wet Tropics, BARPA-R shows a substantially different minimum temperature-precipitation relationship to AGCD and ERA5. This suggests that in the aforementioned regions (Figure 12 f-h). BARPA-R is unable to perform well relative to AGCD and ERA5 and does not reproduce the observed daily minimum temperature-precipitation relationship in DJF season (Figure 12 f-h). BARPA-R performs considerably better at simulating the observed precip-tasmax relationship in

Northern Australia (Figure 11 g, h), resolving the strong seasonal differences between DJF and JJA apparent in the Monsoonal North and Wet Tropics. In these regions, the DJF correlations are strongly negative with maximum values between -0.55 and -0.65.

The maximum strength of the correlations between precipitation and minimum/maximum temperature between the two variables is generally quite strong ($\pm$0.3 - $\pm$0.4) in the NRM clusters for all the datasets. The strongest correlation ($\pm$0.6) between precipitation and maximum temperature is observed in DJF season over the Monsoonal North and Wet Tropics in all the datasets. In all other regions, BARPA-R precipitation is more sensitive to minimum temperature in the summertime (DJF) relative to AGCD and ERA5, hence BARPA-R shows a slightly larger magnitude in correlations (with all peaking about zeroth lag). Further results can be found in the supplementary section where the spatial maps of Spearman's ranked correlation coefficients at lag 0 between the daily precipitation and minimum temperature outputs from 1985-2014 are shown (Figs. A4 and A5), in DJF and JJA in BARPA-R, AGCD and ERA5 datasets.

Overall , BARPA-R simulates realistic relationships between daily maximum temperatures and precipitation across all NRM clusters and in both seasons. Relationships between daily minimum temperatures and precipitation are also well simulated in midlatitude regimes, namely during winter, and across southern Australia. In convective regimes, such as Northern and Central Australia and to a lesser extent along the East Coast, a shift in the minimum temperature-precipitation relationship is apparent in observational datasets but not reflected in BARPA-R. Instead, the BARPA-R JJA relationship persists into DJF in these clusters. These conclusions are consistent with spatial maps of the zero-lag correlations provided in supplementary Figures A4 and A5.

The skilful representation of multivariate relationships has implications for the interpretation of climate risk assessments of compound weather events. For example, hot and dry conditions may lead to enhanced bushfire risk, while hot and humid conditions is associated with enhanced heat stress on humans and livestock. In regions where models struggle to represent correct multivariate relationships, simulations of compound events may be adversely impacted. Improvements to the representation of atmospheric convection, either through improved parametrisation or explicit simulation, may improve the minimum temperature precipitation relationship in the northern Australian wet season.

## 6   Discussion and Conclusions

This paper has analysed the ability of the BARPA-R RCM to maintain a realistic Australian climate when driven with ERA5 reanalysis. Performance in the simulation of Australian temperatures and precipitation was found to be frequently on par with and sometimes improved on the ERA5 reanalysis, despite the contribution of data assimilation in ERA5. This analysis considered mean state biases, seasonality and interannual variability of key ICCLIM metrics chosen to describe the temperature and precipitation climates in the Australian region. Precipitation and temperature teleconnections of the SAM, ENSO and IOD were shown to be well captured by BARPA-R when the appropriate circulation signals are present in the driving boundary inputs and sea surface temperatures. Contemporary change signals of warming were present and, in many cases, overestimated in BARPA-R, while contemporary wetting signals in Northern Australia were underestimated.

Key mean-state biases that exceeded those present in ERA5 included JJA cold biases in daily maximum temperatures of around 1°C across the southern NRM clusters and JJA warm biases in daily minimum temperatures, together leading to a reduced cold season diurnal temperature range. These JJA temperature biases are also evident in MetUM-based regional re-analyses (**?**). The mean monthly maximums in daily precipitation were overestimated by 2-12 mm/day across all NRM clusters in both summer and winter. DJF rain-day counts were improved in northern regions compared to ERA5 but degraded in southern regions. The simulation of near-surface wind speeds was improved compared to ERA5, but nevertheless underestimated the tail of the distribution in all but the two northmost NRM clusters.

BARPA-R shows improvements in mean-state biases over Australia when compared to the previous generation of RCMs, namely CORDEX-CMIP5 (Di Virgilio et al., 2019b) and the ESCI prototype BARPA-R simulations (Su et al., 2021). The pronounced June-August maximum temperature cold bias, which ranged from -2 to -5 °C in CORDEx-CMIP5, is substantially reduced to -1.1 °C in BARPA-R. The mean-state east coast precipitation bias is reduced but remains substantial with an overall DJF mean of 10 mm/day. The bias in the number of overall rain days is reduced in the DJF season from values of up to 5 extra days per month in the ESCI-BARPA simulations to 1-2 extra days across all NRM clusters in BARPA-R. Meanwhile, the ESCI-BARPA underestimation of heavy rain day frequency by 1.5 - 2 days in the wet tropics is transformed to a 0.1 day positive bias in BARPA-R. These changes are likely to be attributable to the inclusion of the improved, 'prognostic entrainment' convection scheme in the new version (Su et al., 2022b).

As BARPA-R projections are intended to produce hazard information for risk assessment purposes, it is important that BARPA-R is able to simulate correct frequencies of hazard-relevant weather and circulation systems. As a first attempt at analysing this, Section 4 focused on the representation of key circulation and large-scale weather systems, such as tropical cyclones, extratropical cyclones, and monsoon westerlies. All circulation and weather systems analysed were present with accurate seasonal cycles in BARPA-R. This is a reassuring but expected result, given that the length-scales of the systems are large, and that these systems are well represented in the driving datasets. Future investigations into the representation of finer scale systems such as sea breeze circulations, drylines, and mountain meteorology may yield more insightful findings. In general, tropical systems such as the monsoon westerlies, tropical cyclones and northwest cloud bands showed larger biases in location and frequency statistics than extra-tropical systems such as extratropical cyclones. These tropical systems also showed less correlation on interannual time-scales than extra-tropical systems. This has implications for future experiment design on hazard analysis. While a case-study approach comparing BARPA-R with its driving model may be appropriate for studying extratropical systems in some instances, it is unlikely to be practical for tropical systems due to divergence between driving and downscaling model behaviour. A larger sample size may therefore be required, especially for studies of rare events such as tropical cyclone landfall.

Both BARPA-R and ERA5 underestimate the intensification trend of wet-day precipitation (SDII) observed in AGCD. This result is consistent with global studies of atmospheric models with parametrised convection, and has been found elsewhere to be rectified by the explicit representation of atmospheric convection (Fowler et al., 2021; Lee et al., 2022; Luu et al., 2022). This is particularly true for subdaily rainfall, which has not been evaluated in this paper. Further downscaling of both climate

projections and regional reanalysis to convection-permitting length-scales over the Australian region are therefore necessary for the assessment of changes in high-intensity, short duration rainfall (Wasko et al., 2023).

Many of the biases and limitations in BARPA-R identified by this study are common biases of the MetUM. These include the overall wet bias (Hudson et al., 2017), the overestimation of the monsoon westerlies (**?**), and the reduced diurnal temperature range in winter (**?**). Future development of BARPA-R will take advantage of on-going MetUM model development, such as the inclusion of the CoMorph convection scheme and updates to the Jules land surface model, with the potential of improving these model shortcomings going forward.

This paper has demonstrated that BARPA-R is able to downscale ERA5 reanalysis to produce a reasonable climate over Australia. This evaluation experiment meets the CORDEX requirement to downscale ERA5 reanalysis in order to evaluate RCM performance in the absence of biased GCM-based driving inputs. Having shown good performance in the evaluation experiment, GCM-based downscaling with BARPA-R is now underway. This BARPA-GCM ensemble will require additional evaluation and is not guaranteed to show similar performance over the Australian region. If key planetary-scale model circulations and processes, such as ENSO or the subtropical jet, are biased or missing in the driving GCM, BARPA-R is unlikely to be able to compensate for these errors. Additionally, nonlinear errors may arise from incompatibility between driving GCMs and the downscaling BARPA-R GCM, such as if the two models have very different favoured vertical profiles of temperature or humidity.

Further work will perform a broader evaluation of BARPA-R's performance at downscaling both ERA5 and CMIP6 GCMs. Benchmarking of the performance of BARPA-R and other CORDEX-CMIP6 RCMs at downscaling historical experiments is needed to establish the credibility of their downscaled projections. The added value of RCMs over GCMs must be evaluated in order to assess the value of computationally expensive dynamical downscaling going forward. Hazard-specific evaluations are required to understand the representation of hazards in BARPA-R simulations before these simulations may be used for risk assessment. Following this evaluation of the full BARPA-R system, these simulations will provide hazards intelligence and climate services to support and inform decision-making.

*Code and data availability.* Processed code and data used in the production of figures in this paper are available at the following DOI: 10.5281/zenodo.8157697. Following submission to CORDEX, the full dataset will be made available via ESGF. Due to intellectual property right restrictions, neither the source code nor documentation papers for the Met Office Unified Model or JULES can be provided directly through open-source repositories. All code used was made available to the editor and reviewers for review.

*Author contributions.* CHS, AJD and CF contributed to the experimental design. SOT provided the initial modelling configuration and advised the model setup CHS, CS, HY and EH set up the model and ran the model simulations. CS, AP, SB and EH post-processed the data and ran feature tracking algorithms. EH designed the evaluation with inputs from all. EH, CHS and RN performed the evaluation. EH led the write-up with inputs and revisions from all.

*Competing interests.* The contact author has declared that none of the authors has any competing interests

*Acknowledgements.* The authors and editor thank Peter Gibson and an anonymous reviewer for their helpful comments on this manuscript. This work was funded by the Australian Climate Service (ACS). We acknowledge valuable inputs from following people: M. Thatcher and M. Dix (CSIRO) on advice on climate modelling; N. Savage, D. Mohit, J. Rodriguez, and L. Kendon (UKMO) for advice on modelling and nudging; I. Bermous (BOM) on UM/JULES optimisation; S. Narsey (BOM), M. Grose (CSIRO), J. Evans (UNSW), and D. Jakob (BOM) for valuable inputs. Valuable and constructive comments by M. Black and U. Bende-Michl are gratefully acknowledged. This work was undertaken with the assistance of resources and services from National Computational Infrastructure (NCI), which is supported by the Australian Government. NCI provides a replication of the ERA5 and ERA5-1 HRES data sets used in this work. ERA5 and ERA5-1 data are produced by ECMWF and distributed via Copernicus Climate Change Service (C3S). As an ESGF node, NCI manages the CMIP collections as well as other ESGF data sets, used in this work.

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

## Appendix A: Supplementary Figures

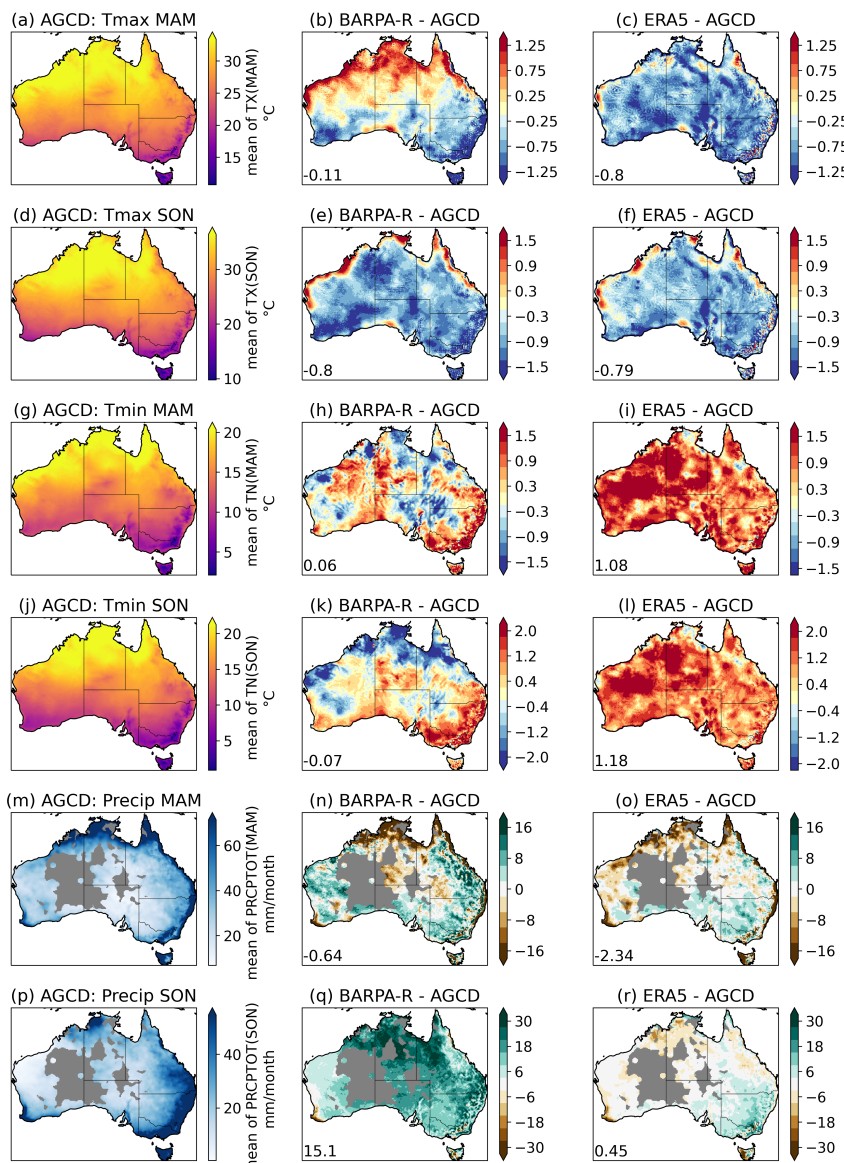

**Figure A1.** Bias in temperature and precipitation climate indicators (rows: TX, TN and PRCPTOT) for transition seasons MAM and SON, for BARPA-R and ERA5 (second and third columns) against AGCD (first column) averaged across the core evaluation period (1985-2014).

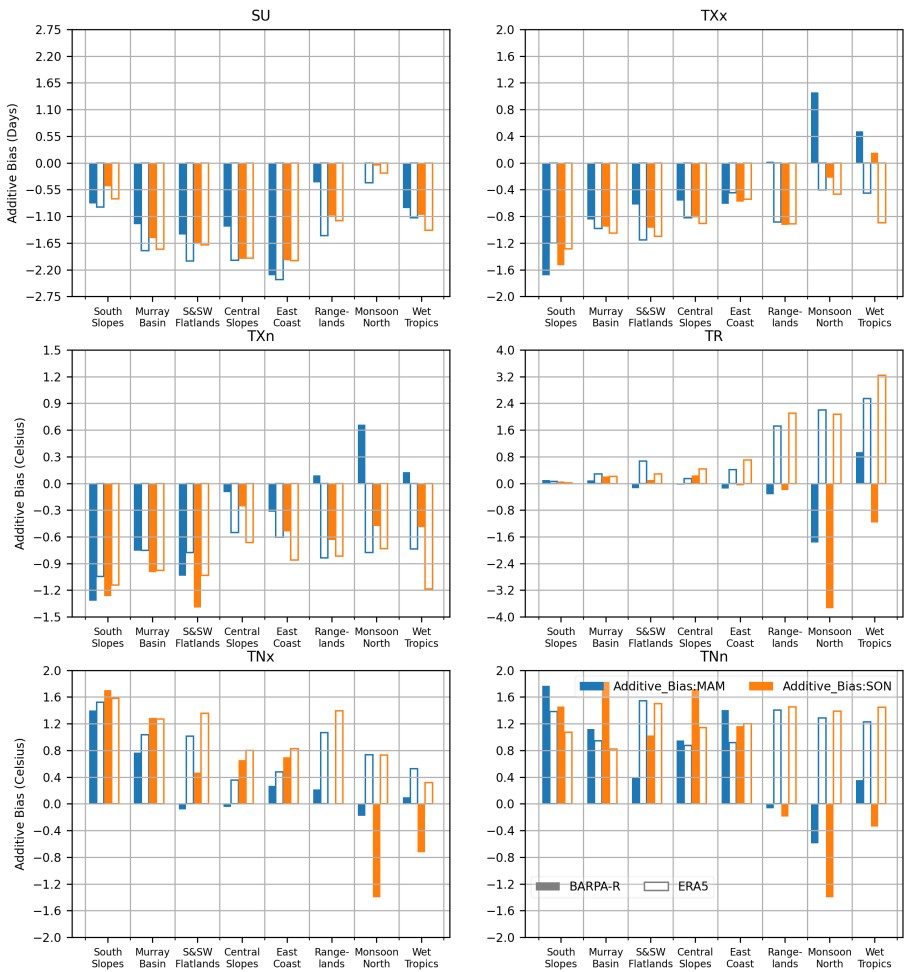

**Figure A2.** BARPA-R (solid bars) and ERA5 (outlined bars) transition season biases of 6 temperature indices across the 8 Australian NRM clusters. Reference data is sourced from AGCD. Panels show number of summer days, (SU; with daily maximum temperatures exceeding 25 ° C), tropical nights, (TN; with daily minimum temperatures exceeding 20 ° C), and the monthly minimums and maximums of the daily minimums and maximums (TNn, TNx, TXn and TXx). Blue and orange bars show the bias aggregated over Austral autumn and spring respectively.

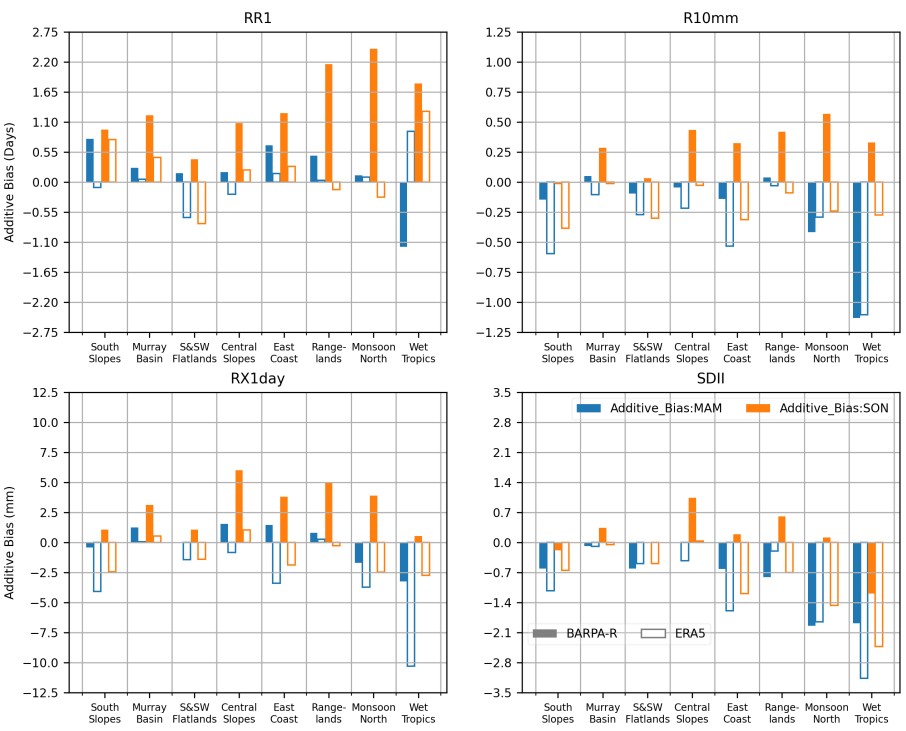

**Figure A3.** As per Figure A2 but for precipitation indices: wet days (RR1; > 1 mm/day), heavy rain days (R10mm; > 10 mm/day), monthly maximum daily precipitation (RX1Day) and the overall monthly precipitation (PRCPTOT).

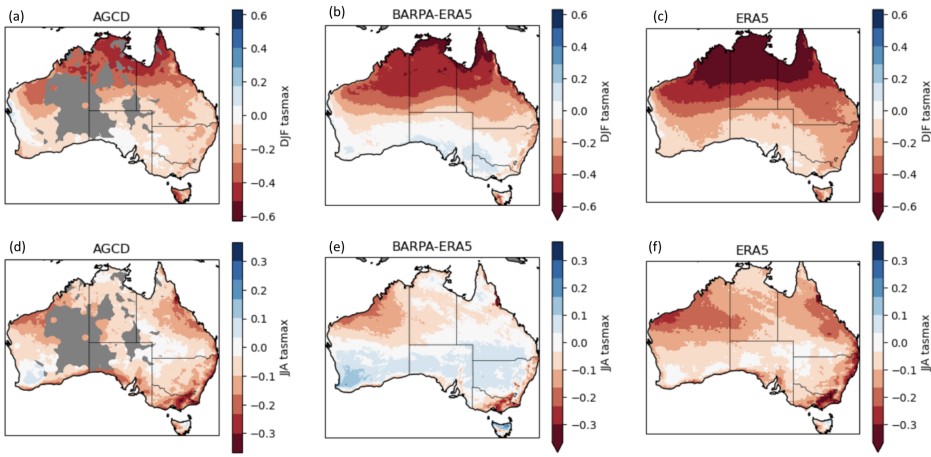

**Figure A4.** Spatial maps of Spearman's ranked correlation coefficients between the daily precipitation and maximum temperature in DJF and JJA in BARPA-ERA5, AGCD and ERA5 datasets.

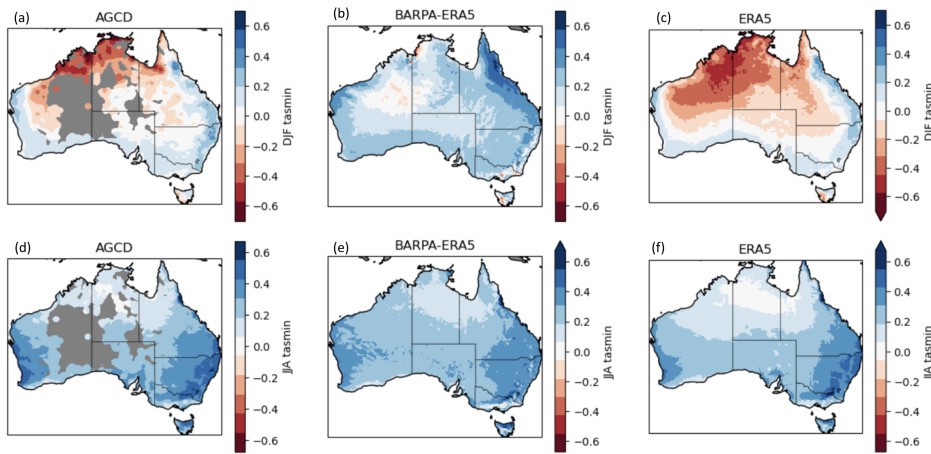

**Figure A5.** Spatial maps of Spearman's ranked correlation coefficients between the daily precipitation and minimum temperature in DJF and JJA in BARPA-ERA5, AGCD and ERA5 datasets.