# Peer review of "Performance and process-based evaluation of the BARPA-R Australasian regional climate model version 1"

_Geoscientific Model Development, 2023_

## Referee Comment (RC1)

**Title:** Review of "Performance and process-based evaluation of the BARPA-R Australasian regional climate model version 1"

Howard et al. (2023)

**Overall Recommendation:** Minor Revisions

**General Comments:**

The paper provides a comprehensive assessment of BARPA-R's performance by downscaling ERA5 reanalysis data for Australia. It examines its representation of various climate variables (surface air temperature, rainfall, and 10-m windspeed), large-scale atmospheric circulation features, and weather systems. Overall, this paper provides a thorough and informative assessment of BARPA-R's performance in downscaling ERA5 for the Australasian region. The findings presented here contribute valuable insights into the model's capabilities and limitations, which will be essential for future climate change research applications.

**Specific Comments:**

**Abstract**

In the abstract, the authors indicate that the temperature assessment is performed in the Kelvin unit (e.g., line 10), but the analysis in the paper seems to be in degree C.

1. **Introduction:**

The paper outlines the need for high-resolution climate projections for hazard risk assessment in Australia. It discusses the need for coordination in high-resolution and accurate climate modelling efforts like CORDEX. They introduce BoM's BARPA-R, a regional climate model for Australasia, in the context of producing high-resolution regional climate projections for Australasia. The paper aims to present an assessment of the BARPA-R simulation using ERA5.

The research question is well-defined and relevant.

- Minor typos:

  - Line 28 – space after "risk".
  - Line 32 – change RCMS to RCMs.

- Line 34 – add McGregor 2005: CCAM: Geometric aspects and dynamical formulation. Technical Paper 70. CSIRO Atmospheric Research, Melbourne.
- The authors should mention and describe "the ESCI prototype" as they utilise it later in the discussion section to show BARPA-R's improvements.

2. **Data and Methods:**

The authors describe in detail the model domain, the experimental design and model configuration, initialisation and boundary conditions, as well as model formulation and coupling. They then introduce the reference dataset (AGCD, ERA5 and BoM's station data) and the evaluation methodology, which focuses on ICCLIM climate indices, seasonal additive

bias, annual multiplicative bias, climatological seasonal correlations and climatological spatial correlations based on NRM regions.

Overall, the "Data and Methods" section and the evaluation approach appear well-structured and comprehensive in their coverage of the research methodology, the reference dataset, and evaluation techniques.

- The authors could provide equations for the indices and metrics calculation.
- Minor typos:
    - Line 91 – "above ground level (AGL)" with capital first letters.
    - Line 131 – Remove the comma before "to a regular grid".

3. **Performance Evaluation:**

   **3.1. Mean State**

   The authors examine the mean-state bias maps of seasonal-mean diurnal maximum and minimum temperatures and precipitation and spatial and temporal characteristics of six temperature and four precipitation indices aggregated over the 8 NRM clusters. Overall, the "Mean State" section offers a thorough and well-structured evaluation of BARPA-R's performance in simulating the climate mean state and key climate metrics.

   - Figure 2: The authors should improve labelling and caption – "TX, TN, and PRCPTOT" labels are misleading. One could assume these are indices, but they are referring to mean Tasmax as TX, mean Tasmin as TN and total precip as PRCPTOT; these need to be explained in the figure caption and in the running text (line 170). Also, indicate in the caption that the digits in the plots represent the averaged bias.
   - Line 174 – The authors refer to a reduced diurnal range, which, I assume, is the difference between maximum and minimum temperature; they could elaborate on how they read this from the plots? Is that the case when Tasmin shows warm bias and Tasmax cold bias over a specific region? This could be explained in the running text; it's a bit confusing to me, and it might not be clear to everyone. Describing the Tasmin and Tasmax patterns separately (and referring to them as daily Tasmax/Tasmin) may be clearer than referring to "diurnal" patterns.
   - Line 183: Is a reference missing "NWP (cite)"?
   - Line 190: I assume with "daily maximums", the authors mean "diurnal maximum"?
   - Line 190: It would be good to have some equations.
   - Line 204-205: It would be good to have some equations.
   - Line 205: Please indicate what SDII stands for and how it is calculated.
   - Figures 3 and 4 – Please improve the caption: indicate what the indices abbreviations stand for, e.g., "number of summer days, **SU,** (with daily maximum temperatures exceeding 25 degrees C)", and so on.
   - Figures 5 and 6 – Add units for "Delta".

**3.2. Trend**

In this chapter, the authors investigate the BARPA-R's ability to simulate climate changes, specifically focusing on its performance in reproducing observed trends in contemporary climate. The analysis presented in this section provides valuable insights into BARPA-R's performance in simulating climate trends. The agreement between BARPA-R and AGCD for temperature-related indices is encouraging, though the discrepancies for minimum temperature-based indices need further investigation. The challenges in simulating extreme rainfall events noted in the warm season are essential findings, and the potential reasons behind this should be explored further, which is probably done in the paper's discussion section?

**3.3. Interannual Variability**

The section focuses on three major modes of interannual climate variability, especially during the Austral Spring, which is highly relevant to understanding the performance of BARPA-R in capturing significant climate drivers in the Australian region. The results suggest that, overall, BARPA-R performs well in simulating the teleconnections associated with ENSO, IOD, and SAM.

The authors could consider exploring the sensitivity of BARPA-R to different model configurations or parameters, which may help improve its performance in capturing interannual temperature variability associated with ENSO.

- Line 251 – what is meant by "diurnal maximum temperatures"? Is it daily Tasmax, or the range between Tasmax and Tasmin? Maybe the authors could avoid using the term "diurnal", as it might confuse some.

**3.4. 10-meter Winds**

The section comprehensively evaluates BARPA-R's performance in simulating 10-metre wind speeds. The improvement in high percentile wind speeds compared to ERA5 (reference) is a positive finding, indicating that BARPA-R better captures extreme wind events in many regions. However, the persistent underestimation of calm weather conditions suggests that there is still room for refinement. The authors could explore whether model parameterisation or resolution adjustments can address this issue?

- Line 262 – remove space between 1 and 0 "of 1 0m AGL".
- Figure 8. – Is it possible to make this plot larger? The labels are hard to read.

**4. Process Evaluation:**
**4.1. Circulation**
This section shows that the four identified large-scale circulation features are present in both BARPA-R and ERA5 (reference), and their seasonal cycles match. Identifying biases and discussing their potential causes contribute to a better understanding of model limitations

and areas for improvement. How can the biases related to the monsoon westerlies and the SPCZ be addressed?

- Line 301 – Remove TODO "(not shown, TODO)"?
- Figure 9 – Change the labels "barpa" and "era5" to upper case. And add colour bar labels.
- Tabel 1 – Captions for tables are placed above the table.

**4.2. Weather Systems**

The authors investigate BARPA-R's performance in simulating key large-scale weather systems that influence Australia, including tropical and extra-tropical cyclones and Australian Northwest Cloud-Bands (NWCBs). While extra-tropical cyclones are generally well represented, biases and decoupling of interannual variability are observed for tropical cyclones and NWCBs. Addressing these issues and further investigating the causes would enhance the overall understanding of BARPA-R's capabilities in simulating important weather systems in the Australasian context.

- Line 325 – What are these direct observational products? I think they are listed in the caption of Figure 10, but they need to be mentioned here as well.
- Line 325 – The authors say that Obs are replacing ERA5, but as they mention in the caption of Figure 10, ERA5 is a part of "Obs". Please clarify.
- Line 339 – The first letters in upper case "outgoing longwave radiation (OLR)"
- Line 344 – Please mention that ERA5 is part of "Obs"? Because you say in line 325 that Obs are replacing ERA5.
- Tabel 2 – Captions for tables are placed above the table.
- Figure 10 – Please add labels to the colour bars.
- Figure 10 – This part of the captions is not clear: "NWCBs (red lines; contour interval: **1 event/season, first contour: 2** events/season)"

**5. Lagged Temperature-Precipitation Relationship:**

In this section, the authors calculate Spearman's ranked correlations with a lag time of ±10 days between diurnal temperature and precipitation outputs, focusing on DJF and JJA over different NRM clusters, and demonstrate that BARPA-R's capability to capture multivariate relationships, particularly between temperature and precipitation, as compared to observational and reanalysis datasets. However, there are regional variations that may have implications for climate impact assessments. How could these discrepancies be addressed? And what are the practical implications of these findings for climate impact assessments?

- Line 370 – Maybe use the term daily temperature-precipitation relationship instead of "diurnal"; it can be confusing.
- Figure 11 – Please add a legend.
- Figure 12 – Caption typo: minimum temperature instead of maximum temperature
- Line 376 – Add the abbreviation "precip" after rainfall, as you start using it in the paragraph below.

- Line 392 – Remove the dot after "(Figure 12 f-h)."
- Line 398 – Remove "between the two variables", it's a repeat.

**6. Discussion and Conclusion:**

Overall, the discussion provides a comprehensive summary of the key findings and implications of the study. It effectively communicates the strengths and weaknesses of BARPA-R, highlights improvements over previous RCMs, and underscores the importance of hazard analysis and ongoing work in downscaling CMIP6 GCMs. The conclusions drawn are well-supported by the study's results and contribute valuable insights to the field of regional climate modelling in Australia. The conclusion highlights the significance of the study and suggests avenues for future research.

The authors could consider providing a more detailed discussion of the potential reasons behind BARPA-R's limitations, and the specific discrepancies observed between BARPA-R and the reference data and recommend potential strategies to address these limitations, e.g., exploring the sensitivity of BARPA-R to different model configurations or parameters.

- Line 423 – Why is the cold bias referred to as "1K" when the analysis was in degrees Celsius?
- Line 429 – The authors mention the ESCI prototype for the first time here. This needs to be referenced and briefly described in the introduction, too.
- Line 431 – Add "C" after -1.1 degrees.

**7. Citations and References:**

The references are appropriate.

**Recommendation:**

I recommend that this journal paper be accepted for publication after addressing the minor revisions suggested above.

---

## Referee Comment (RC2)

Review of "Performance and process-based evaluation of the BARPA-R Australasian regional climate model version 1" for GMD

**General comments**

The authors introduce BARPA-R, a regional model used for downscaling reanalysis and GCMs at ~17km resolution over Australia. The paper focuses first on evaluation of the ERA5 reanalysis driven simulations, this is useful for isolating regional model biases. The evaluation is very comprehensive and extends far beyond the standard climatological metrics, to include aspects of large-scale atmospheric circulation and feature tracking, as well as lagged metrics to consider land atmosphere feedbacks.

The authors have been careful in their experimental design to align with CORDEX requirements. The model and experimental design appropriately described and justified.

My comments below are mostly minor and relate to improving the presentation and discussion of certain results. Overall, I think this is an important well-written and comprehensive manuscript.

**Specific comments**

-should be "ERA5" not "ERA-5" check for consistency throughout

-Line 138, what version of ERA5 was used? Given the known issues with lower stratospheric temperature and humidity biases https://www.ecmwf.int/en/elibrary/81149-global-stratospheric-temperature-bias-and-other-stratospheric-aspects-era5-and . I doubt this matters much for downscaling, but useful to document

-Line 95, include additional details about what atmospheric variables from ERA5 are used to drive BARPA, and at what levels etc.

-Is AGCD the same as AWAP? Perhaps discuss further. Related to this, are there papers that have tried to quantify the observational uncertainty from this product? This is likely important for some biases. See additional comments below.

-Lines 140 onwards, I agree with the idea that similar biases in BARPA-R and ERA5 should be interpreted as good (or at least acceptable) for the reasons discussed. But I think this needs to be highlighted in the abstract and/or conclusion – as it is fundamental to the results presented here and this point could be missed by readers.

-Lines 155, suggest explaining additive and multiplicative biases more thoroughly

-Figure 2 (and others). The number in the panel, what is this exactly? Is it the mean bias? This should be detailed in the figure or caption. It would be useful to include MAE (mean absolute bias) also, so that this isn't contaminated by cancellation of errors of different signs which clearly is present in some of the results. At other times you present additive and multiplicative biases so I was confused.

-Figure 3 and 4 – this is very comprehensive – but there is a lot packed into these figures. I wonder if the number of comparisons can be reduced (different bars) so that the figure is easier to read (or broken into a separate figure). The 3 separate y-axis seems somewhat excessive in my opinion.

-Figure 5 – Nice figure. Further discussion around what might be contributing to these differences in trends seems warranted. For temperature, does ERA5 assimilate observations in a way that is temporally consistent (i.e. so that local trends are expected to be realistic)?

-Figure 6, can you be sure that AGCD is appropriate to use for trend analysis of rainfall – i.e. considering station inhomogeneity and the effects of interpolation? For example, the following paper found highly inconsistent trends for precipitation indices over the US in observational products (including different in situ gridded products):

Gibson, P. B., D. E. Waliser, H. Lee, B. Tian, and E. Massoud, 2019: Climate Model Evaluation in the Presence of Observational Uncertainty: Precipitation Indices over the Contiguous United States. J. Hydrometeor., 20, 1339–1357, https://doi.org/10.1175/JHM-D-18-0230.1.

-Lines 280, Suggested discussion point - has the BARPA-R wet bias been shown to be a general bias seen in other UM regional models at this approximate resolution?

-Line 301- there is a "not shown, TODO" left in that needs updating :)

-Figure 9-10. Given the BARPA-R is forced by ERA5, it would be quite odd if large scale features like the climatology of the subtropical jet diverged much. So, the agreement is not that surprising. Smaller scale convective features are where we would expect more divergence, as you show. Perhaps worth discussing this point more.

-Figure 11. This is very interesting and well presented – a valuable contribution to the paper

---

## Author Response (AR1)

We thank the reviewers for taking the time to review our manuscript and for their constructive comments on our paper. We respond to each point below, where the reviewer comments are in regular text and our responses are in blue.

**Reviewer 1**

**Abstract**

In the abstract, the authors indicate that the temperature assessment is performed in the Kelvin unit (e.g., line 10), but the analysis in the paper seems to be in degree C.

Corrected

1. **Introduction:**

   The research question is well-defined and relevant.

   - Minor typos:

        o Line 28 – space after "risk".

        Corrected

        o Line 32 – change RCMS to RCMs.

        Corrected

   - Line 34 – add McGregor 2005: CCAM: Geometric aspects and dynamical formulation. Technical Paper 70. CSIRO Atmospheric Research, Melbourne.
     Added
   - The authors should mention and describe "the ESCI prototype" as they utilise it later in the discussion section to show BARPA-R's improvements.
     Added

2. **Data and Methods:**
   - The authors could provide equations for the indices and metrics calculation.
     We have added the equations for the metrics near line 180
   - Minor typos:
        o Line 91 – "above ground level (AGL)" with capital first letters.
        Corrected
        o Line 131 – Remove the comma before "to a regular grid".
        Corrected

3. **Performance Evaluation:**

   ### 3.1. Mean State

   - Figure 2: The authors should improve labelling and caption – "TX, TN, and PRCPTOT" labels are misleading. One could assume these are indices, but they are referring to mean Tasmax as TX, mean Tasmin as TN and total precip as

PRCPTOT; these need to be explained in the figure caption and in the running text (line 170). Also, indicate in the caption that the digits in the plots represent the averaged bias.

corrected

- Line 174 – The authors refer to a reduced diurnal range, which, I assume, is the difference between maximum and minimum temperature; they could elaborate on how they read this from the plots? Is that the case when Tasmin shows warm bias and Tasmax cold bias over a specific region? This could be explained in the running text; it's a bit confusing to me, and it might not be clear to everyone. Describing the Tasmin and Tasmax patterns separately (and referring to them as daily Tasmax/Tasmin) may be clearer than referring to "diurnal" patterns.

We have added the following text immediately prior to this discussion: "When temperature biases show a decrease in maximum temperatures coupled to an increase in minimum temperatures in the same season, this can be interpreted as an underestimation of the diurnal temperature range."

- Line 183: Is a reference missing "NWP (cite)"?

Added Hudson et al 2017

- Line 190: I assume with "daily maximums", the authors mean "diurnal maximum"?

Removed all references to tasmax and tasmin as 'diurnal max/min' following your later recommendation

- Line 190: It would be good to have some equations.

We have added the equations for the metrics near line 180

- Line 204-205: It would be good to have some equations.

We have added the equations for the metrics near line 180

- Line 205: Please indicate what SDII stands for and how it is calculated.

Clarified to read: "SDII - the Simple Daily (precipitation) Intensity Index, which is calculated as the average precipitation rate across all days with at least 1 mm of precipitation."

- Figures 3 and 4 – Please improve the caption: indicate what the indices abbreviations stand for, e.g., "number of summer days, SU, (with daily maximum temperatures exceeding 25 degrees C)", and so on.

Corrected

- Figures 5 and 6 – Add units for "Delta".

Added

**3.2. Trend**

The agreement between BARPA-R and AGCD for temperature-related indices is encouraging, though the discrepancies for minimum temperature-based indices need further investigation. The challenges in simulating extreme rainfall events noted in the warm season are essential findings, and the potential reasons behind this should be explored further, which is probably done in the paper's discussion section?

We have added the text to the discussion: both BARPA-R and ERA5 underestimate the intensification trend of wet-day precipitation (SDII) observed in AGCD. This result is

consistent with global studies of atmospheric models which parameterise convection, and has been found to be rectified by the explicit representation of atmospheric convection (Fowler et al 2021, Lee et al 2022, Luu et al 2020). This is particularly true for sub-daily extreme rainfall, which has not been evaluated in this paper. Further downscaling of climate projections and regional reanalysis to convection permitting length-scales over the Australian region are therefore necessary for the assessment of changes in high intensity, short duration rainfall (Wasko et al, 2023).

**3.3. Interannual Variability**

The section focuses on three major modes of interannual climate variability, especially during the Austral Spring, which is highly relevant to understanding the performance of BARPA-R in capturing significant climate drivers in the Australian region. The results suggest that, overall, BARPA-R performs well in simulating the teleconnections associated with ENSO, IOD, and SAM.

The authors could consider exploring the sensitivity of BARPA-R to different model configurations or parameters, which may help improve its performance in capturing interannual temperature variability associated with ENSO.

Unfortunately, this suggested analysis is beyond the scope of this paper. We expect the response to ENSO to be highly dependant on the driving models in future GCM downscaling experiments.

- Line 251 – what is meant by "diurnal maximum temperatures"? Is it daily Tasmax, or the range between Tasmax and Tasmin? Maybe the authors could avoid using the term "diurnal", as it might confuse some.

    Removed all references to tasmax and tasmin as 'diurnal max/min'. We have retained the term 'diurnal temperature range' to refer to tasmax-tasmin in section 3.1

**3.4. 10-meter Winds**

The section comprehensively evaluates BARPA-R's performance in simulating 10-metre wind speeds. The improvement in high percentile wind speeds compared to ERA5 (reference) is a positive finding, indicating that BARPA-R better captures extreme wind events in many regions. However, the persistent underestimation of calm weather conditions suggests that there is still room for refinement. The authors could explore whether model parameterisation or resolution adjustments can address this issue?

This is an instrumental issue: the stations do not measure wind speeds between 0 and 2 m/s and instead record these low wind speeds as 0 m/s. To avoid confusion, we have removed these low percentiles from figure 8.

- Line 262 – remove space between 1 and 0 "of 1 0m AGL".
  Corrected
- Figure 8. – Is it possible to make this plot larger? The labels are hard to read.

4. **Process Evaluation:**

   **4.1. Circulation**

This section shows that the four identified large-scale circulation features are present in both BARPA-R and ERA5 (reference), and their seasonal cycles match. Identifying biases and discussing their potential causes contribute to a better understanding of model limitations and areas for improvement. How can the biases related to the monsoon westerlies and the SPCZ be addressed?

This paper aims to document the key biases and limitations of BARPA-R. Resolving these biases would require considerable model development and is currently out of scope. Ocean coupling may help with the SPCZ biases.

We have added the text:

This westerly wind bias has been linked by Martin et al 2019 to errors in the representation of convection and precipitation over the Maritime Continent and the western/central equatorial Indian Ocean. Systematic rainfall biases in the maritime continent region are common due to the complex, multiscale nature of convection in this region.

A reduced southerly bias in the location of the SPCZ has been documented in the ACCESS-S1 seasonal forecast system (Beischer et al 2021), suggesting that ocean coupling may improve the representation of the SPCZ.

- Line 301 – Remove TODO "(not shown, TODO)"?
  Corrected
- Figure 9 – Change the labels "barpa" and "era5" to upper case. And add colour bar labels.
  done
- Tabel 1 – Captions for tables are placed above the table.
  Corrected

   **4.2. Weather Systems**

The authors investigate BARPA-R's performance in simulating key large-scale weather systems that influence Australia, including tropical and extra-tropical cyclones and Australian Northwest Cloud-Bands (NWCBs). While extra-tropical cyclones are generally well represented, biases and decoupling of interannual variability are observed for tropical cyclones and NWCBs. Addressing these issues and further investigating the causes would enhance the overall understanding of BARPA-R's capabilities in simulating important weather systems in the Australasian context.

- Line 325 – What are these direct observational products? I think they are listed in the caption of Figure 10, but they need to be mentioned here as well.

  This information is given in the next paragraph. This text has been improved as follows for clarity:

> "In this analysis, some direct observational products are available, and these are used as references where possible. For weather systems where no direct observation is available, ERA5 is used as the reference. Identification algorithms and reference datasets are described below."

- Line 325 – The authors say that Obs are replacing ERA5, but as they mention in the caption of Figure 10, ERA5 is a part of "Obs". Please clarify.

  Please see correction above. ERA5 is used where no direct observation is available.

- Line 339 – The first letters in upper case "outgoing longwave radiation (OLR)"

  Corrected

- Line 344 – Please mention that ERA5 is part of "Obs"? Because you say in line 325 that Obs are replacing ERA5.

  Replaced 'ERA5' with 'the ERA5-based reference'

- Tabel 2 – Captions for tables are placed above the table.

  Corrected

- Figure 10 – Please add labels to the colour bars.

  done

- Figure 10 – This part of the captions is not clear: "NWCBs (red lines; contour interval: **1 event/season, first contour: 2** events/season)"

  corrected

5. **Lagged Temperature-Precipitation Relationship:**

In this section, the authors calculate Spearman's ranked correlations with a lag time of ±10 days between diurnal temperature and precipitation outputs, focusing on DJF and JJA over different NRM clusters, and demonstrate that BARPA-R's capability to capture multivariate relationships, particularly between temperature and precipitation, as compared to observational and reanalysis datasets. However, there are regional variations that may have implications for climate impact assessments.

How could these discrepancies be addressed? And what are the practical implications of these findings for climate impact assessments?

We have added the text

The skilful representation of the relationship between temperature and precipitation has implications for the interpretation of climate risk assessments of compound weather events. For example, hot and dry conditions may lead enhanced bushfire risk, while hot and humid conditions can lead to enhanced heat stress on humans and livestock. In regions where models struggle to represent the correct temperature-precipitation relationship, simulation of compound events may be adversely affected. Improvements to the representation of atmospheric convection, either through improved parametrisation or explicit simulation, may improve the minimum temperature – precipitation relationship in the northern Australian wet season.

Line 370 – Maybe use the term daily temperature-precipitation relationship instead of "diurnal"; it can be confusing.

*Removed all references to diurnal*

- Figure 11 – Please add a legend.

  *todo*

- Figure 12 – Caption typo: minimum temperature instead of maximum temperature

  *Corrected*

- Line 376 – Add the abbreviation "precip" after rainfall, as you start using it in the paragraph below.

  *Replaced each occurrence of 'precip' and 'rainfall' with 'precipitation'*

- Line 392 – Remove the dot after "(Figure 12 f-h)."

  *This is meant to be there: end of a sentence*

- Line 398 – Remove "between the two variables", it's a repeat.

  *corrected*

**6. Discussion and Conclusion:**

Overall, the discussion provides a comprehensive summary of the key findings and implications of the study. It effectively communicates the strengths and weaknesses of BARPA-R, highlights improvements over previous RCMs, and underscores the importance of hazard analysis and ongoing work in downscaling CMIP6 GCMs. The conclusions drawn are well-supported by the study's results and contribute valuable insights to the field of regional climate modelling in Australia. The conclusion highlights the significance of the study and suggests avenues for future research.

The authors could consider providing a more detailed discussion of the potential reasons behind BARPA-R's limitations, and the specific discrepancies observed between BARPA-R and the reference data and recommend potential strategies to address these limitations, e.g., exploring the sensitivity of BARPA-R to different model configurations or parameters.

*We have added the text:*

*Many of the biases and limitations in BARPA-R identified by this study are common biases of the MetUM. These include the overall wet bias (Hudson et al 2017), the overestimation of the monsoon westerlies (Martin et al 2021) and the reduced diurnal temperature range during winter (Su et al 2023), Future development of BARPA-R will take advantage of the ongoing MetUM model development, such as the inclusion of the CoMorph convection scheme and updates to the Jules land surface model, with the potential of improving model shortcomings going forward.*

- Line 423 – Why is the cold bias referred to as "1K" when the analysis was in degrees Celsius?
  *Corrected*

- Line 429 – The authors mention the ESCI prototype for the first time here. This needs to be referenced and briefly described in the introduction, too.

  *Text added as follows in the introduction: "This model is a continuation of prototype work developed for the Energy Sector Climate Information (ESCI) project, documented by \cite{Su2019} and hereon referred to as ESCI-BARPA."*

- Line 431 – Add "C" after -1.1 degrees.
  *Corrected*

**7. Citations and References:**

The references are appropriate.

**Reviewer 2: Peter Gibson**

**Specific comments**

-should be "ERA5" not "ERA-5" check for consistency throughout

Corrected

-Line 138, what version of ERA5 was used? Given the known issues with lower stratospheric temperature and humidity biases https://www.ecmwf.int/en/elibrary/81149-global-stratospherichttps://www.ecmwf.int/en/elibrary/81149-global-stratospheric-temperature-bias-and-other-stratospheric-aspects-era5-andtemperature-bias-and-other-stratospheric-aspects-era5-and . I doubt this matters much for downscaling, but useful to document

We did use ERA5.1: we've added the text " Between 2000 and 2006, boundary inputs were derived from ERA5.1 to avoid stratospheric temperature and humidity biases present in the original ERA5 dataset."

-Line 95, include additional details about what atmospheric variables from ERA5 are used to drive BARPA, and at what levels etc.

We have updated this to read: "Boundary conditions were updated every 3-hours and derived from the ERA5 pressure level dataset, which consists of 37 vertical levels. The 3D model inputs from ERA5 at the lateral boundaries were horizontal winds, specific humidity, temperature, cloud liquid, cloud ice and cloud cover."

-Is AGCD the same as AWAP? Perhaps discuss further. Related to this, are there papers that have tried to quantify the observational uncertainty from this product? This is likely important for some biases. See additional comments below.

Yes, AGCDv1 is AWAP, we've now indicated this. We've also added following the text describing observational uncertainty in AGCD.

"Jones et al 2009 describe key sources of observational uncertainty in AGCD. They highlight underestimations of maximum temperatures in regions of tight climate gradients and sparse observational coverage, including the coastal north-west Australia and the Nullarbor Plain due to poor resolution of maritime effects. They also note large analysis errors in daily rainfall estimates, with mean absolute errors up to 50% of the total. King et al. (2013) demonstrated that AGCD is suitable for use in studies of trends, extremes and variability in rainfall across much of Australia, with limitations occurring in regions where station coverage is sparse. Meanwhile, Chubb et al (2016) established large systematic dry biases between AGCD and an independent gauge network in the snowy mountains. In the following analysis, the direction of the AGCD biases is opposite to the BARPA-R bias presented. This means that the biases presented in this paper are likely overestimates, ensuring that our analysis is conservative."

-Lines 140 onwards, I agree with the idea that similar biases in BARPA-R and ERA5 should be interpreted as good (or at least acceptable) for the reasons discussed. But I think this needs to be

highlighted in the abstract and/or conclusion – as it is fundamental to the results presented here and this point could be missed by readers.

Added the following text to the abstract: "Performance-based evaluation results are benchmarked against ERA5, with comparable performance between the free-running BARPA-R simulations and observationally constrained reanalysis interpreted as a good result."

-Lines 155, suggest explaining additive and multiplicative biases more thoroughly

We have added equations near line 180. We have also changed the text to refer to additive bias as 'bias' and multiplicative bias as 'variance error' for clarity.

-Figure 2 (and others). The number in the panel, what is this exactly? Is it the mean bias? This should be detailed in the figure or caption. It would be useful to include MAE (mean absolute bias) also, so that this isn't contaminated by cancellation of errors of different signs which clearly is present in some of the results. At other times you present additive and multiplicative biases so I was confused.

We have added this text to the caption and added the MAE.

-Figure 3 and 4 – this is very comprehensive – but there is a lot packed into these figures. I wonder if the number of comparisons can be reduced (different bars) so that the figure is easier to read (or broken into a separate figure). The 3 separate y-axis seems somewhat excessive in my opinion.

We have split each panel into 3 panels

-Figure 5 and 6 – Nice figure. Further discussion around what might be contributing to these differences in trends seems warranted. For temperature, does ERA5 assimilate observations in a way that is temporally consistent (i.e. so that local trends are expected to be realistic)? Can you be sure that AGCD is appropriate to use for trend analysis of rainfall – i.e. considering station inhomogeneity and the effects of interpolation? For example, the following paper found highly inconsistent trends for precipitation indices over the US in observational products (including different in situ gridded products): Gibson, P. B., D. E. Waliser, H. Lee, B. Tian, and E. Massoud, 2019: Climate Model Evaluation in the Presence of Observational Uncertainty: Precipitation Indices over the Contiguous United States. J. Hydrometeor., 20, 1339–1357, https://doi.org/10.1175/JHM-D-18-0230.1.

ERA-5 does have issues with long-term temperature trends as you suggest. Jones et al 2009 goes into considerable detail on how AGCD was designed to be robust for long-term trends. We have included the following text.

"This trend analysis must be caveated by the observational uncertainties associated with the trends of both AGCD and ERA5. Long-term trends in observational datasets, including analyses and reanalyses are sensitive to temporal inhomogeneities in their input datasets (e.g. Gibson 2019). Simmons et al (2021) found that temperature trends over Australia are affected by inhomogeneities in the observational inputs, however the poorest performance occurs prior to 1970, before our study period. AGCD has been designed to be more robust to long-term trends, through the application of an anomaly-based approach which takes advantage of climate normals at a subset of stations with longer coverage. Jones et al 2009 demonstrate that this approach provides consistent maps of rainfall trends compared to monthly analyses derived only from stations with long climate records and found that temperature trends were similarly robust at the large scale.

With these caveats in mind, this paper accounts for observational uncertainty in rainfall trends by focussing attention on established trends that have been studied elsewhere, namely southern Australian cool season drying, wetting trends in north-western Australian during summer, and the intensification of short-duration heavy convective rainfall.

-Lines 280, Suggested discussion point - has the BARPA-R wet bias been shown to be a general bias seen in other UM regional models at this approximate resolution?

There is a persistent wet bias over Australia in different UM models, as is discussed briefly on line 183. However literature on climate-mode regional UM simulations over Australia is naturally limited.

-Line 301- there is a "not shown, TODO" left in that needs updating :)
Corrected

-Figure 9-10. Given the BARPA-R is forced by ERA5, it would be quite odd if large scale features like the climatology of the subtropical jet diverged much. So, the agreement is not that surprising. Smaller scale convective features are where we would expect more divergence, as you show. Perhaps worth discussing this point more.
We have added the following text to the discussion section:
All circulation and weather systems analysed were present with accurate seasonal cycles in BARPA-R. This is a reassuring but expected result, given that the length-scales of the systems are large, and that these systems are well represented in the driving datasets. Future investigations into the representation of finer scale systems such as sea breeze circulations, drylines, and mountain meteorology may yield more insightful findings.

-Figure 11. This is very interesting and well presented – a valuable contribution to the paper
Thanks Peter!